# Lactate dehydrogenase D is a general dehydrogenase for D-2-hydroxyacids and is associated with D-lactic acidosis

Shan Jin[1,3], Xingchen Chen [1,3], Jun Yang[1] & Jianping Ding [1,2] ✉

Mammalian lactate dehydrogenase D (LDHD) catalyzes the oxidation of D-lactate to pyruvate. LDHD mutations identified in patients with D-lactic acidosis lead to deficient LDHD activity. Here, we perform a systematic biochemical study of mouse LDHD (mLDHD) and determine the crystal structures of mLDHD in FAD-bound form and in complexes with FAD, $Mn^{2+}$ and a series of substrates or products. We demonstrate that mLDHD is an $Mn^{2+}$-dependent general dehydrogenase which exhibits catalytic activity for D-lactate and other D-2-hydroxyacids containing hydrophobic moieties, but no activity for their L-isomers or D-2-hydroxyacids containing hydrophilic moieties. The substrate-binding site contains a positively charged pocket to bind the common glycolate moiety and a hydrophobic pocket with some elasticity to bind the varied hydrophobic moieties of substrates. The structural and biochemical data together reveal the molecular basis for the substrate specificity and catalytic mechanism of LDHD, and the functional roles of mutations in the pathogenesis of D-lactic acidosis.

Lactate (LAC) is an important metabolite, which exists as two enantiomers due to the presence of a chiral C2 atom, namely L-lactate and D-lactate. L-lactate mainly comes from the reduction of pyruvate (PYR) during the final step of anaerobic glycolysis[1,2], whereas D-lactate is produced in miniscule amounts either endogenously through methylglyoxal metabolism pathway or exogenously by intestinal bacterial metabolism and food consumption[3,4]. Under normal physiological conditions, the concentration of L-lactate in circulation is much higher than that of D-lactate[5–8]. Accumulation of excess L- and D-lactate in the human body can cause lactic acidosis[9]. L-lactic acidosis is a common clinical problem, while D-lactic acidosis is a relatively rare disease[10,11]. D-lactic acidosis is commonly considered as a complication of short-bowel syndrome, which occurs after removal of part of the small intestine due to malignant tumors, diseases, or jejunoileal bypass surgery[11–15]. Shortening of the small intestine impairs the absorption of carbohydrates, resulting in increased transmission of carbohydrates to colon bacteria, which could proliferate and form an acidic

environment conducive to D-lactate production. Although both L-lactate and D-lactate can pass through the blood-brain barrier, unlike L-lactate, accumulation of D-lactate in brain has neurotoxic effects, leading to diverse neurological symptoms, including speech disorder, gait disturbance, and changes in mental state and behavior, which are often confused with primary nervous system diseases[10–12]. The onset and severity of D-lactic acidosis are still not well understood.

The reversible conversion of pyruvate to L-lactate is catalyzed by a family of intracellular NAD-dependent enzymes called L-lactate dehydrogenases (L-LDHs)[1,2,16,17]. In mammals, there are three types of L-LDH isoenzymes, namely LDHA, LDHB, and LDHC, which form homo- or hetero-tetramers with differed tissue specificities and varied efficacies[17–20]. Malfunction of L-LDHs can lead to abnormal accumulation of L-lactate, which is associated with various diseases including L-lactic acidosis and cancers[1,2,9,16]. High serum L-LDH levels are typically related to poor prognosis in many types of cancers and thus are considered as an indicator of tumor burden and metastasis as well as a

[1]State Key Laboratory of Molecular Biology, Shanghai Institute of Biochemistry and Cell Biology, Center for Excellence in Molecular Cell Science, University of Chinese Academy of Sciences, Chinese Academy of Sciences, 320 Yue-Yang Road, Shanghai 200031, China. [2]School of Life Science and Technology, ShanghaiTech University, 393 Middle Huaxia Road, Shanghai 201210, China. [3]These authors contributed equally: Shan Jin, Xingchen Chen. ✉e-mail: jpding@sibcb.ac.cn

complex biomarker pertinent to the activation of several oncogenic signaling pathways[1,2,16,21,22]. So far, the structures, functions and catalytic mechanisms of LDHA, LDHB, and LDHC have been well elucidated and their relationships to various diseases especially cancers are under active scrutiny[1,2,16,17].

D-lactate can be oxidized to pyruvate by D-lactate dehydrogenase (D-LDH). There are two types of D-LDHs: NAD-dependent D-LDHs and FAD-dependent D-LDHs[23,24]. The NAD-dependent D-LDHs were found in some bacteria and belong to the D-isomer specific 2-hydroxyacid dehydrogenase superfamily; however, the exact biological functions remain elusive[25–27]. The FAD-dependent D-LDHs were initially found in bacteria, archaea, yeasts and plants[28–38]. These enzymes are evolutionarily unrelated to L-LDHs and NAD-dependent D-LDHs, and belong to the 2-hydroxyacid dehydrogenase subfamily of the VAO/PCMH flavoprotein family, all members of which consist of a conserved FAD-binding domain and a variable substrate-binding domain, and use FAD rather than NAD as cofactor[23,24,39]. Besides D-LDH, this subfamily also includes D-2-hydroxyglutarate dehydrogenase (D2HGDH) and bacterial glycolate oxidase subunit D (GlcD), both of which catalyze the oxidation of the 2-hydroxyl group of substrate to a carbonyl group as well[23,24,39,40].

Recently, FAD-dependent D-LDHs were also identified in mammals including human and mouse, which are localized in the mitochondrial inner membrane and share high sequence similarity with yeast D-LDH (DLD1, Supplementary Fig. 1)[8,41,42]. Human D-LDH (also called LDHD) is highly expressed in tissues with high metabolic rates and abundant mitochondria[43]. Very recently, several missense mutations of human LDHD were found in patients either with neurological symptoms, including global growth retardation, cerebellar ataxia, and transient hepatomegaly[8,44], or with autosomal recessive gout and hyperuricemia[43]. Analyses of the metabolic profiles show that these patients have substantially elevated concentrations of D-lactate and several other organic acids including D-2-hydroxyisovalerate (D-2-HIV), D-2-hydroxyisocaproate (D-2-HIC), 3-hydroxybuyrate (3-HB), and 2-hydroxy-3-methylvalerate (2-HMV) in urine and plasma[8,44]. These disease-associated mutations were suggested to render loss-of-function LDHD, leading to increased levels of D-lactate and other organic acids in urine and plasma.

So far, the structure, function and molecular basis for the substrate specificity and catalytic reaction of LDHD are unknown. The functional roles of LDHD mutations in the pathogeneses of D-lactic acidosis remain elusive. In this work, we carry out a systematic biochemical study of the enzymatic properties of mouse LDHD (mLDHD) and show that mLDHD exhibits enzymatic activity for D-lactate but no activity for L-lactate. In addition, mLDHD exhibits enzymatic activities towards a broad range of D-2-hydroxyacids with hydrophobic moieties including D-2-hydroxyisovalerate, D-2-hydroxyisocaproate and D-2-hydroxy-3-methylvalerate, but no activity for these with hydrophilic moieties such as D-malate and D-2-hydroxyglutarate. We determine the crystal structures of mLDHD in FAD-bound form and in complexes with FAD, $Mn^{2+}$ and a series of D-2-hydroxyacid substrates or their products. The functional roles of the key residues at the active site and the disease-associated mutations were validated by mutagenesis and enzymatic assays. Our structural and biochemical data together demonstrate that LDHD is an $Mn^{2+}$-dependent general dehydrogenase for a broad range of D-2-hydroxyacids containing hydrophobic moieties and thus may play an important role in the metabolism of these organic acids. In addition, our data reveal the molecular basis for the substrate specificity and catalytic mechanism of LDHD, and for the functional roles of disease-associated mutations in the pathogenesis of D-lactic acidosis.

## Results
### Expression and purification of mouse LDHD
Human and mouse LDHDs share high sequence identity (81.5%) and similarity (92.6%). The major difference is that human LDHD (hLDHD) contains an insertion of 23 residues (residues 210–232) compared to mouse LDHD (mLDHD) (Supplementary Fig. 1). We initially tried to express hLDHD in *E. coli* cells; however, various attempts yielded only insoluble inclusion bodies. However, we could express an N-terminal truncated mLDHD (residues 22–484) in *E. coli* and purified the recombinant protein with high purity and homogeneity as examined by size-exclusion chromatography and SDS-PAGE analyses (Supplementary Fig. 2). LDHD belongs to the 2-hydroxyacid dehydrogenase subfamily of the VAO/PCMH flavoprotein family, all members of which require FAD as cofactor[23,24,39]. The purified mLDHD protein exhibits a characteristic yellow color of FAD with maximum absorbance at 450 nm, and exists as a monomer in solution (Supplementary Fig. 2a). Quantification of the FAD and protein concentrations shows that the molar ratio of FAD:protein is $0.73 \pm 0.04$ (mean ± SEM) in the wild-type (WT) mLDHD protein. To remove any potential unknown divalent metal ions bound to mLDHD derived from the expression and purification processes, we added 1 mM EDTA in the lysis buffer and storage buffer. Later structure determination of mLDHD in FAD-bound form confirms that there is no metal ion bound at the active site (see discussion later).

### mLDHD has activity for D-2-hydroxyacids with hydrophobic moieties
To characterize the enzymatic properties of mLDHD, we first analyzed the effects of different divalent metal ions on the activity to catalyze the conversion of D-lactate to pyruvate. As expected, the EDTA-treated mLDHD has no activity in the absence of metal ions (Fig. 1a). Among the eight common divalent metal ions examined ($Mn^{2+}$, $Mg^{2+}$, $Ni^{2+}$, $Co^{2+}$, $Zn^{2+}$, $Cd^{2+}$, $Cu^{2+}$, and $Ca^{2+}$), mLDHD exhibits the highest activity in the presence of $Mn^{2+}$, and moderate to weak activity in the presence of $Co^{2+}$, $Ni^{2+}$ and $Ca^{2+}$ (about 49%, 16% and 11% of the activity compared to $Mn^{2+}$, respectively), but no measurable activity in the presence of $Mg^{2+}$, $Zn^{2+}$, $Cd^{2+}$, and $Cu^{2+}$ (Fig. 1a). Intriguingly, $Zn^{2+}$ appears to affect the stability of mLDHD as addition of $Zn^{2+}$ immediately abolishes the yellow color of FAD. It is evident that the divalent cation serves as a cofactor but not an activator for mLDHD. Next, we examined the dependence of the activity on the $Mn^{2+}$ concentration and the results show that mLDHD exhibits optimal specific activity with the $Mn^{2+}$ concentration in the range of 30–200 µM (Supplementary Fig. 3). Then, we examined the dependence of the activity on the pH of the reaction solution and the results show that mLDHD displays the maximum specific activity at pH 7.4 (Fig. 1b), which is in agreement with the weak alkaline physiological environment in vivo. Thus, we used a buffer of pH 7.4 containing 50 µM $Mn^{2+}$ in all of the following activity assay.

At the standard assay conditions, mLDHD exhibits enzymatic activity with a catalytic efficiency ($k_{cat}/K_M$) of 0.50 min$^{-1}$·µM$^{-1}$ to catalyze the conversion of D-lactate into pyruvate with a $V_{max}$ of $1.22 \pm 0.01$ µmol·min$^{-1}$·mg$^{-1}$, a $K_M$ of $122.0 \pm 4.5$ µM and a $k_{cat}$ of $61.0 \pm 0.6$ min$^{-1}$ (the kinetic parameters are presented as the mean ± SEM; Fig. 1c, Table 1). Due to the rapid catalytic reaction, we could not reliably measure the binding of D-lactate with mLDHD using ITC, SPR or other methods. On the other hand, mLDHD exhibits no activity for L-lactate and has no detectable binding with L-lactate. These results indicate that mLDHD has high stereo-selectivity for D-lactate over L-lactate.

Previous studies have shown that in addition to D-lactate, increased levels of D-2-hydroxyisovalerate, D-2-hydroxyisocaproate, 2-hydroxy-3-methylvalerate, and 3-hydroxybuyrate are also found in urine and plasma of patients harboring LDHD mutants, suggesting that LDHD may be involved in the regulation of these hydroxyacids[8,44]. Similar to D-lactate, D-2-hydroxyisovalerate, D-2-hydroxyisocaproate and D-2-hydroxy-3-methylvalerate all contain a D-2-glycolate moiety (including the C1-carboxyl and C2-hydroxyl groups) and a hydrophobic moiety at the D-chiral C2 atom; however, D-3-hydroxybuyrate

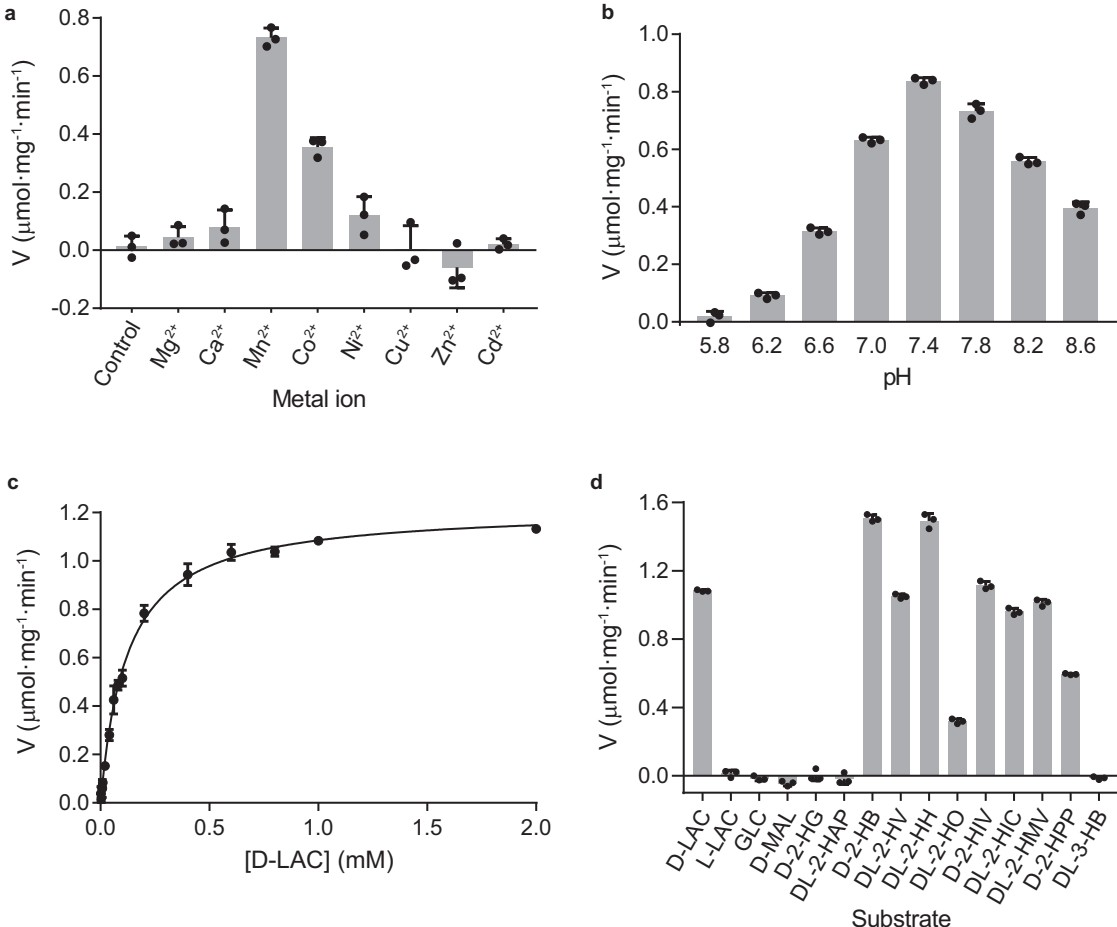

**Fig. 1 | Enzymatic properties of mLDHD. a** The specific activity of mLDHD to convert D-lactate (D-LAC) to pyruvate (PYR) in the absence and presence of different divalent metal ions (1 mM). **b** The specific activity of mLDHD towards D-LAC under different pH. **c** The saturation curve of mLDHD towards D-LAC. **d** The specific activity of mLDHD towards different substrates and analogs (1 mM). For the racemic substrates, the substrate concentration used in the experiment was doubled and the specific activity was calculated based on the corrected concentration of D-isomer (1/2 of the concentration of racemic substrate). Data are presented as the mean ± SEM ($n$ = 3 independent measurements). Source data are provided as a Source Data file. D-LAC D-lactate, L-LAC L-lactate, GLC glycolate, D-MAL D-malate, D-2-HG D-2-hydroxyglutarate, DL-2-HAP DL-2-hydroxy-3-aminopropionate, D-2-HB D-2-hydroxybutyrate, DL-2-HV DL-2-hydroxyvalerate, DL-2-HH DL-2-hydroxyhexanoate, DL-2-HO DL-2-hydroxyocatanoate, D-2-HIV D-2-hydroxyisovalerate, DL-2-HIC DL-2-hydroxycaproate, DL-2-HMV DL-2-hydroxy-3-methylvalerate, D-2-HPP D-2-hydroxy-3-phenylpropionate, and DL-3-HB DL-3-hydroxybutyrate.

contains a C3-hydroxyl group instead of a C2-hydroxyl group (Supplementary Fig. 4). Thus, we examined whether mLDHD has activity towards these hydroxyacids. The results show that mLDHD can convert D-2-hydroxyisovalerate, D-2-hydroxyisocaproate and D-2-hydroxy-3-methylvalerate into 2-ketoisovalerate, 2-ketoisocaproate, and 2-keto-3-methylvalerate, respectively, with comparable $V_{max}$ and $k_{cat}$ values and significantly decreased $K_M$ values compared to those for D-lactate, resulting in substantially increased catalytic efficiencies (5.5-8.0 folds); however, mLDHD shows no activity towards D-3-hydroxybuyrate (Fig. 1d and Table 1).

The above results inspired us to further examine whether mLDHD has activity towards other D-2-hydroxyacids with longer or larger hydrophobic moieties at the C2 atom (Supplementary Fig. 4). The results show that mLDHD also exhibits activity towards D-2-hydroxybutyrate, D-2-hydroxyvalerate and D-2-hydroxyhexanoate with comparable $V_{max}$ and $k_{cat}$ values and moderately decreased $K_M$ values compared to these for D-lactate, resulting in moderately increased catalytic efficiency (2.5–4.2 folds) (Fig. 1d and Table 1). However, mLDHD exhibits 2.5-fold decreased activity with substantially decreased $V_{max}$ and $k_{cat}$ values and comparable $K_M$ value towards D-2-hydroxyoctanoate which contains a longer aliphatic moiety, and 4.3-fold decreased activity with slightly decreased $V_{max}$ and $k_{cat}$ values and

moderately increased $K_M$ value towards D-2-hydroxy-3-phenylpropionate which contains a larger methylenephenyl moiety (Fig. 1d and Table 1).

Moreover, we examined whether mLDHD has activity towards glycolate and D-2-hydroxyacids with hydrophilic moieties attached to the C2 atom such as D-2-hydroxy-3-amino-propanoate, D-malate and D-2-hydroxyglutarate (Supplementary Fig. 4). The results show that mLDHD has no activity towards these hydroxyacids (Fig. 1d). Consistently, mLDHD shows no detectable binding with these hydroxyacids.

It is worth noting that although the DL-isomer mixtures for some substrates (DL-2-hydroxyvalerate, DL-2-hydroxyhexanoate, DL-2-hydroxyoctanoate, DL-2-hydroxyisocaproate, and DL-2-hydroxy-3-methylvalerate) were used in the activity assay, mLDHD displays comparable activity and kinetic parameters compared to the pure D-isomer substrates with similar chemical structures (Fig. 1d and Table 1), indicating that the presence of L-isomers has no impact on the activity towards the D-isomers and implying that the L-isomers cannot bind to mLDHD. These results are in agreement with the biochemical data showing that the activity of mLDHD towards D-lactate was not affected in the presence of L-lactate, 3-hydroxybutyrate, D-malate, and D-2-hydroxyglutarate (Supplementary Fig. 5).

**Table 1 | Kinetic parameters of mLDHD towards different substrates[a]**

| Substrates | Products | $V_{max}$ (µmol·min$^{-1}$·mg$^{-1}$) | $K_M$ (µM) | $k_{cat}$ (min$^{-1}$) | $k_{cat}/K_M$ (min$^{-1}$·µM$^{-1}$) |
|---|---|---|---|---|---|
| D-lactate (D-LAC) | Pyruvate (PYR) | 1.22 ± 0.01 | 122.0 ± 4.5 | 61.0 ± 0.6 | 0.50 |
| D-2-hydroxybutyrate (D-2-HB) | 2-ketobutyrate (2-KB) | 1.60 ± 0.02 | 62.9 ± 2.8 | 80.0 ± 0.9 | 1.27 |
| DL-2-hydroxyvalerate (DL-2-HV)[b] | 2-ketovalerate (2-KV) | 1.12 ± 0.02 | 31.4 ± 2.7 | 56.0 ± 1.1 | 1.78 |
| DL-2-hydroxyhexanoate (DL-2-HH) | 2-ketohexanoate (2-KH) | 1.48 ± 0.02 | 34.9 ± 1.6 | 74.0 ± 0.7 | 2.12 |
| DL-2-hydroxyoctanoate (DL-2-HO) | 2-ketooctanoate (2-KO) | 0.409 ± 0.010 | 104.0 ± 9.0 | 20.5 ± 0.5 | 0.197 |
| D-2-hydroxyisovalerate (D-2-HIV) | 2-ketoisovalerate (2-KIV) | 1.11 ± 0.02 | 13.9 ± 1.2 | 55.5 ± 0.9 | 3.99 |
| DL-2-hydroxyisocaproate (DL-2-HIC) | 2-ketoisocaproate (2-KIC) | 0.902 ± 0.017 | 16.4 ± 0.7 | 45.1 ± 0.8 | 2.75 |
| DL-2-hydroxy-3-methyl-valerate (DL-2-HMV) | 2-keto-3-methylvalerate (2-KMV) | 1.01 ± 0.01 | 16.5 ± 0.1 | 50.5 ± 0.6 | 3.06 |
| D-2-hydroxy-3-phenyl-propionate (D-2-HPP) | 2-keto-3-phenylpropionate (2-KPP) | 0.788 ± 0.037 | 337.0 ± 44.0 | 39.4 ± 1.8 | 0.117 |

[a]The $V_{max}$, $K_M$ and $k_{cat}$ values of mLDHD were determined at the standard reaction conditions with varied concentrations of substrate, and are presented as mean ± SEM ($n$ = 3 independent experiments).
[b]For the racemic substrates, the substrate concentration used in the experiment was doubled, and the kinetic parameters were calculated based on the corrected concentration of D-isomer (1/2 of the concentration of racemic substrate).

Compared to other metabolic enzymes, the $k_{cat}$ and $k_{cat}/K_M$ values of mLDHD are comparable to those of secondary metabolic enzymes that are involved in the regulation of metabolites present in specific cells or tissues, under specific circumstances and/or at relatively low levels, but are lower than those of central metabolic enzymes that are involved in the main carbon and energy flow and the metabolism of amino acids, fatty acids and nucleotides[45]. This indicates that mLDHD is a secondary metabolic enzyme, which is in agreement with the high tissue-specific expression of mLDHD[43] and the miniscule amount of D-lactate (and very likely other D-2-hydroxyacids) in the human body under normal physiological conditions[3].

Taken together, our biochemical data demonstrate that mLDHD is an Mn$^{2+}$-dependent general dehydrogenase for a broad range of D-2-hydroxyacids with small to moderate-size hydrophobic moieties at the C2 atom. However, it has no activity for glycolate or D-2-hydroxyacids with hydrophilic moieties at the C2 atom. In addition, mLDHD exhibits high stereo-selectivity for the D-isomers over the L-isomers.

**Crystal structures of mLDHD**

To elucidate the molecular basis for the substrate preference and catalytic reaction of mLDHD, we determined the crystal structure of wild-type (WT) mLDHD in FAD-bound form (mLDHD$^{FAD}$, 1.70 Å resolution) and in complex with FAD, Mn$^{2+}$, and D-lactate (mLDHD$^{FAD+Mn+D-LAC}$, 1.73 Å resolution) (Supplementary Data 1). Crystallization experiments show that soaking of the mLDHD$^{FAD}$ crystals in crystallization solution supplemented with the substrates and MnCl$_2$ often yields the product-bound or mixed substrate/product-bound structures. Thus, we prepared the loss-of-function H405A mutant mLDHD (mLDHD$_{H405A}$), and determined the crystal structures of the mLDHD$_{H405A}$ mutant in FAD-bound form and in complexes with FAD and various substrates (D-lactate, D-2-hydroxybutyrate, D-2-hydroxyvalerate, D-2-hydroxyhexanoic acid, D-2-hydroxyoctanoate, D-2-hydroxyisovalerate, D-2-hydroxyisocaproate, and D-2-hydroxy-3-methylvalerate) at high resolutions (1.31–1.75 Å) (Supplementary Data 1). In addition, we also determined the crystal structures of WT mLDHD in complexes with Mn$^{2+}$ and a number of products (pyruvate, 2-ketobutyrate, 2-ketovalerate, 2-ketohexanoate, 2-ketoisovalerate, 2-ketoisocaproate, and 2-keto-3-methylvalerate) at high resolutions (1.55–2.00 Å) (Supplementary Data 1).

In all structures, there is one mLDHD molecule in the asymmetric unit, and most residues of mLDHD are well-defined in the electron density maps except for a few surface exposed residues. The disordered surface residues are omitted from the final structure models (Supplementary Data 1). There is a non-covalently bound FAD at the active site, which is apparently derived from the expression system, indicating that FAD binds to mLDHD tightly. This is consistent with the

observation that the purified protein displays a characteristic yellow color of FAD. In the WT mLDHD$^{FAD}$ and mLDHD$_{H405A}$$^{FAD}$ structures, there is no density for metal ion and ligand at the active site (Supplementary Fig. 6a, b). In the substrate-bound mLDHD and mLDHD$_{H405A}$ structures, all the substrates are clearly defined in the electron density maps; however, Mn$^{2+}$ is only identified at the active site in the WT structure, but not in the mutant structures due to the mutation of the metal-coordinating His405 (Supplementary Fig. 6c–k). In the product-bound WT mLDHD structures, all the products and Mn$^{2+}$ are well defined in the electron density maps (Supplementary Fig. 6l–r). The bound substrates and products can be distinguished unambiguously based on the electron density and the chirality of the C2 atom: the substrates contain a D-chiral C2 atom and the C1, C3, and O3 atoms are not coplanar, whereas the products contain a non-chiral C2 atom and the C1, C3, and O3 atoms are coplanar (Supplementary Fig. 6s). Although the DL-isomer mixtures of some substrates (DL-2-hydroxyvalerate, DL-2-hydroxyhexanoate, DL-2-hydroxyoctanoate, DL-2-hydroxyisocaproate, and DL-2-hydroxy-3-methylvalerate) were used in the crystallization experiments, only the D-isomers bind to the active site, further confirming that mLDHD has high stereo-selectivity for the D-isomers over the L-isomers.

Similar to other members of the VAO/PCMH flavoprotein family[39,46,47], mLDHD consists of an FAD-binding domain, a substrate-binding domain and a C-terminal domain (Fig. 2a). The FAD-binding domain (residues 27–243) adopts a PCMH-type FAD-binding domain which is comprised of two subdomains: subdomain A (residues 27-119) contains a three-strand parallel β-sheet (β1–β3) surrounded by three α-helices (α1–α3), and subdomain B (residues 120–243) contains a four-strand antiparallel β-sheet (β4–β7) surrounded by two α-helices (α4 and α5). The substrate-binding domain (residues 244–446) is comprised of a seven-strand reverse parallel β-sheet (β8–β14) wrapped by six α-helices (α6–α11). The small C-terminal domain (residues 447–484) consists of two α-helices (α12–α13), which covers over the FAD-binding site.

The active site resides at the interface of the FAD-binding domain and the substrate-binding domain, and is composed of the FAD-binding site and the substrate-binding site (Fig. 2b). There is a channel connecting the substrate-binding site to the solvent, which is formed by both hydrophilic and hydrophobic residues, including Glu58, Thr99, Gly159, Ala160, His348, Asn349, and Tyr352 (Fig. 2b). This channel appears to be the transporting path for the substrate into and the product out of the active site and thus is designated as the "substrate-loading channel".

Like in the structures of other VAO/PCMH flavoprotein family members, the FAD-binding site is located at the interface of the two subdomains of the FAD-binding domain and is covered by the

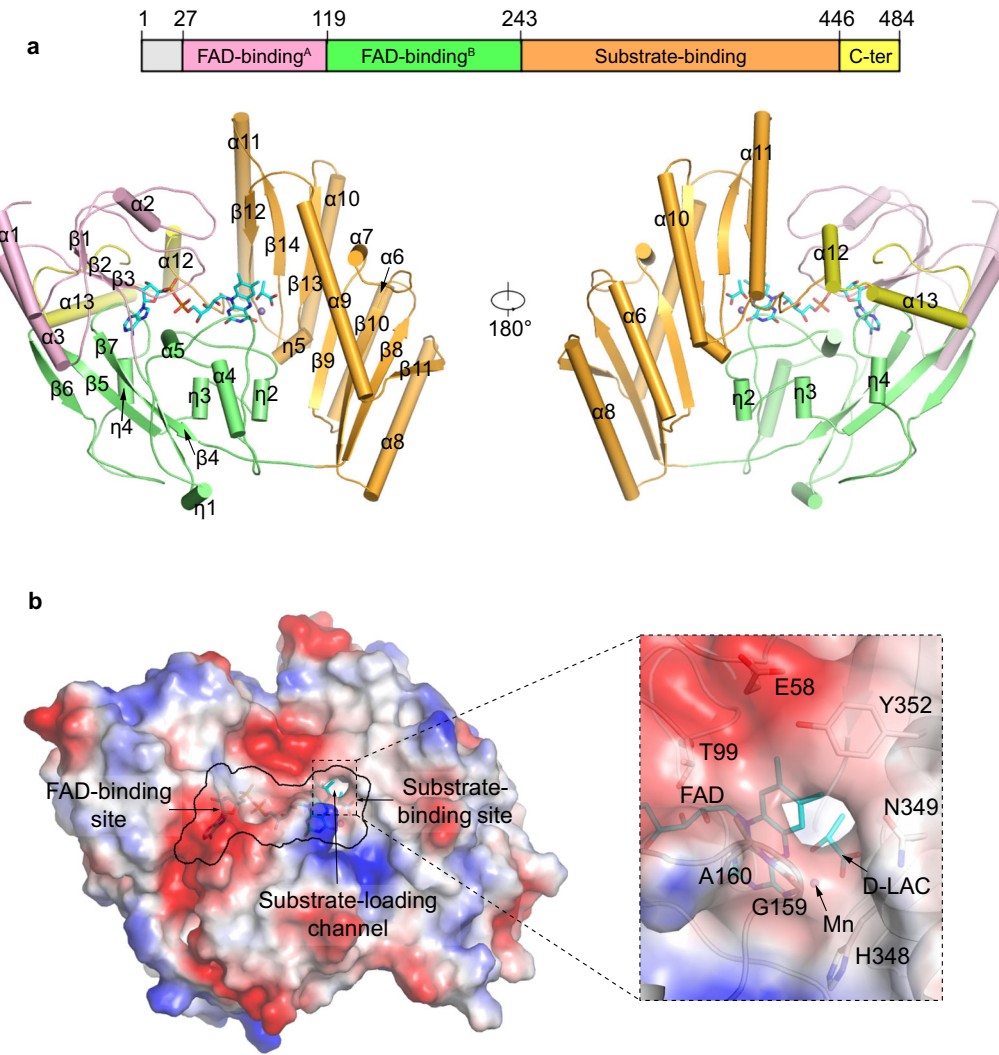

**Fig. 2 | Crystal structure of mLDHD. a** Overall structure of mLDHD bound with FAD, Mn²⁺ and D-lactate (D-LAC) with cartoon presentation in two different orientations. The FAD-binding subdomains A and B are colored in pink and green, the substrate-binding domain in orange, and the C-terminal domain in yellow, respectively. FAD and D-LAC are shown in cyan stick models, and Mn²⁺ as a gray sphere. **b** Electrostatic surface of mLDHD showing the locations of the FAD-binding site, the substrate-binding site, and the substrate-loading channel. The zoom-in panel shows the composition of the substrate-loading channel with the key residues shown in stick models.

C-terminal domain (Fig. 2a). The bound FAD adopts an extended conformation with the adenosine diphosphate (ADP) and ribitol moieties buried in the deep inside of the FAD-binding site and the flavin moiety positioned at the interface of the FAD-binding domain and the substrate-binding domain (Fig. 2a). The FAD forms extensive hydrophilic and hydrophobic interactions with the surrounding residues (Supplementary Fig. 7). Besides, in all the Mn²⁺ and ligand-bound structures, the O4 atom of the flavin moiety forms a coordination bond with Mn²⁺. Sequence alignment shows that most residues involved in the FAD binding are highly conserved in LDHD homologs from other eukaryotes, underscoring their functional role in the FAD binding (Supplementary Fig. 1).

### Structure of the substrate-binding site

The substrate-binding site is located on the *si*-face of the flavin moiety of FAD and consists of two subsites (Fig. 3a, b). Subsite A is a positively charged pocket composed of conserved hydrophilic residues including Arg347, His398, His405, Glu442, and His443, which binds Mn²⁺ and the glycolate moiety (the C1-carboxyl and C2-hydroxyl groups) of substrate or the glyoxylate moiety (the C1-carboxyl and C2-keto groups) of product. Subsite B is a hydrophobic pocket composed of

Val101, Trp351, Ser365, and Ile407, which binds the hydrophobic moiety of substrate or product.

In the mLDHD^FAD structure, the substrate-binding site is occupied by a water molecule (Supplementary Fig. 6a). In the mLDHD^FAD+Mn+D-LAC structure, the substrate-binding site is bound with an Mn²⁺ and a D-lactate (Supplementary Fig. 6c). In this structure, Mn²⁺ forms six coordination bonds in an octahedral geometry with the side chains of His398, His405, and Glu442, the O4 atom of the flavin moiety of FAD, and the C1-carboxyl and C2-hydroxyl groups of D-lactate (Fig. 3b). In addition to its interactions with Mn²⁺, the C1-carboxyl group of D-lactate also forms two hydrogen bonds with the sidechain of Arg347. However, the C3 atom makes only a few hydrophobic interactions with the surrounding residues. Comparison of the mLDHD^FAD and mLDHD^FAD+Mn+D-LAC structures shows that the binding of Mn²⁺ and D-lactate causes only very subtle conformational changes of a few residues composing subsite A including His398, His405, and Glu442, which rotate their side chains slightly to better coordinate Mn²⁺ (Supplementary Fig. 8a).

Comparison of the mLDHD^FAD and mLDHD_H405A^FAD structures reveals no notable conformational differences at the active site (Supplementary Fig. 8b). Comparison of the mLDHD^FAD+Mn+D-LAC and

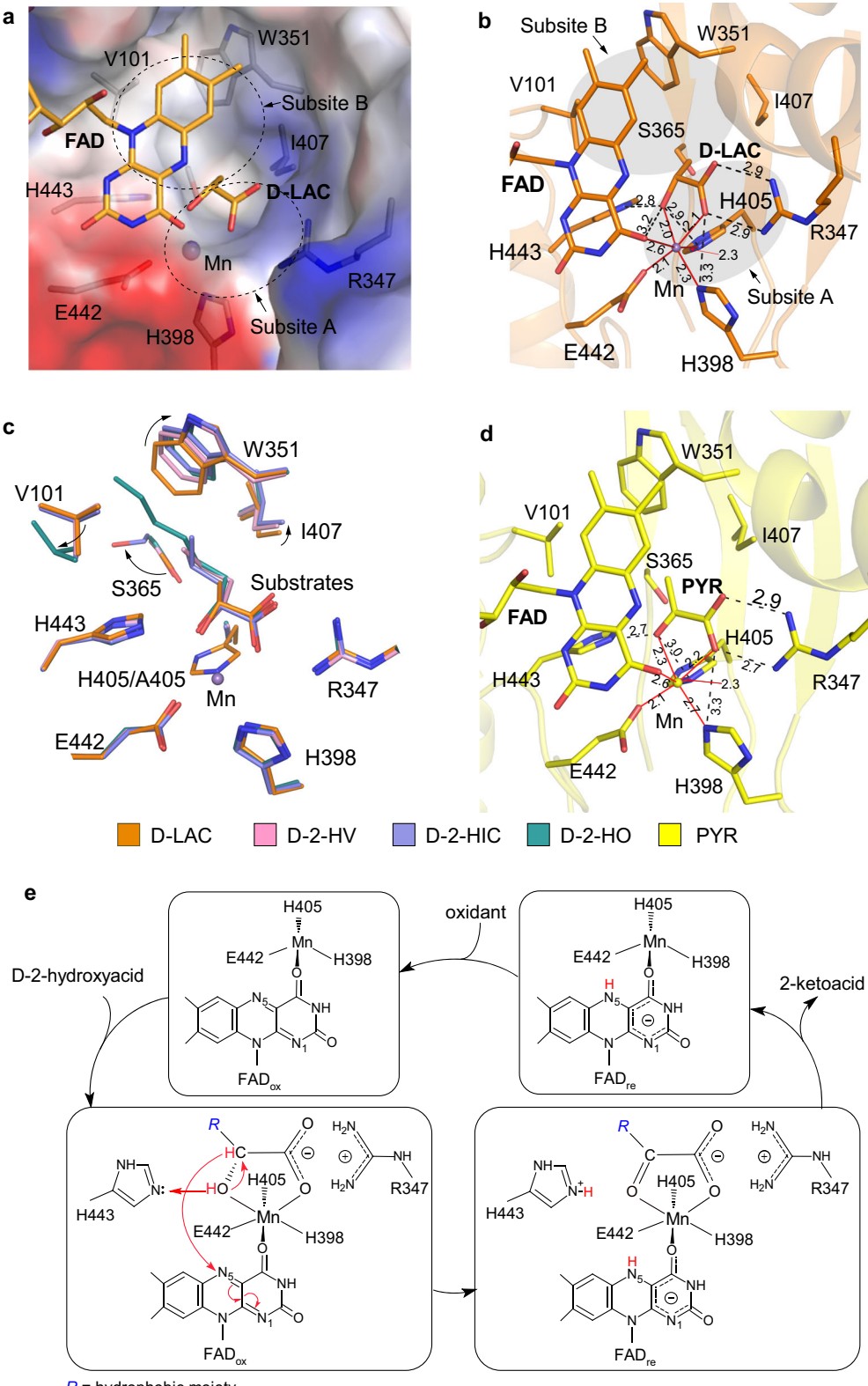

**Fig. 3 | Structure of the substrate-binding site of mLDHD. a** Electrostatic surface of the substrate-binding site in the mLDHD$^{FAD+Mn+D-LAC}$ structure. The substrate-binding site consists of the subsite A of hydrophilic property and the subsite B of hydrophobic property. **b** Interactions between D-lactate (D-LAC), FAD, Mn$^{2+}$, and the surrounding residues in the mLDHD$^{FAD+Mn+D-LAC}$ structure. The hydrogen-bonding interactions are shown with black dashed lines, and the coordination bonds with Mn$^{2+}$ are shown with red solid lines. The lengths (Å) of hydrogen bonds and coordination bonds are indicated. **c** Superposition of the substrate-binding sites in the representative D-lactate (D-LAC), D-2-hydroxyvalerate (D-2-HV), D-2-hydroxyisocaproate (D-2-HIC), and D-2-hydroxyoctanoate (D-2-HO)-bound mLDHD structures showing the conformational changes of the key residues forming the substrate-binding subsite B to varied extents along with the increase of the size of the hydrophobic moiety attached to the C2 atom of the substrate. **d** Hydrogen-bonding interactions between pyruvate (PYR), FAD, Mn$^{2+}$, and the surrounding residues in the mLDHD$^{FAD+Mn+PYR}$ structure. **e** Catalytic mechanism of mLDHD. *R* represents the hydrophobic moiety at the D-chiral C2 atom of the substrate.

mLDHD$_{H405A}$$^{FAD+D\text{-}LAC}$ structures shows that in the mutant mLDHD$_{H405A}$ structure, D-lactate binds to the active site with a slightly different orientation due to the absence of Mn$^{2+}$; nevertheless, the residues composing the active site exhibit no major conformational changes and D-lactate maintains similar interactions with the surrounding residues as in the WT mLDHD structure (Supplementary Fig. 8c). Comparison of the D-lactate-bound and other substrate-bound mLDHD$_{H405A}$ structures also reveals no major conformational changes for the residues at the substrate-binding subsite A, and the other substrates bind to the same site and the C1-carboxyl and C2-hydroxyl groups of substrates maintain similar hydrogen-bonding interactions with the surrounding residues as D-lactate (Supplementary Fig. 8d). However, along with the increase of the size of the hydrophobic moiety of substrate, the key residues composing the substrate-binding subsite B undergo conformational changes to varied extents to accommodate the larger hydrophobic moiety (Fig. 3c and Supplementary Fig. 8d). Concurrently, the larger hydrophobic moiety of substrate makes more hydrophobic interactions with the surrounding residues.

In all the product-bound (pyruvate, 2-ketobutyrate, 2-ketovalerate, 2-ketohexanoate, 2-ketoisovalerate, 2-ketoisocaproate, and 2-keto-3-methylvalerate) mLDHD structures, Mn$^{2+}$ maintains similar coordination bonds with the surrounding residues, FAD, and the C1-carboxyl and C2-keto groups of product as that in the mLDHD$^{FAD+Mn+D\text{-}LAC}$ structure (Fig. 3d and Supplementary Fig. 8e). In addition, the C1-carboxyl group of product maintains similar hydrogen-bonding interactions with the sidechain of Arg347 as that of D-lactate; however, the C2-keto group makes no hydrogen-bonding interaction with the sidechain of Glu442 (Fig. 3d and Supplementary Fig. 8e). Moreover, the residues composing the substrate-binding subsite B in the product-bound structures adopt similar conformations as those in the corresponding substrate-bound structures and undergo similar conformational changes to varied extents along with the increase of the size of the hydrophobic moiety of product (Supplementary Fig. 8f).

## Molecular basis for the substrate specificity

Our biochemical data demonstrate that mLDHD has activity for a range of D-2-hydroxyacids containing small to moderate-size hydrophobic moieties, but no activity for their L-isomers, glycolate, or D-2-hydroxyacids containing hydrophilic moieties. Analysis of the substrate-bound mLDHD and mLDHD$_{H405A}$ structures provides the molecular basis for the substrate specificity of mLDHD. The substrate-binding site of mLDHD consists of a positively charged pocket (subsite A) to bind Mn$^{2+}$ and the glycolate moiety of substrate and a hydrophobic pocket (subsite B) to bind the hydrophobic moiety of substrate. In addition to the hydrophilic interactions of the glycolate moiety with the protein at the subsite A, the hydrophobic moiety also makes favorable hydrophobic interactions with the protein at the subsite B, further stabilizing the substrate binding. This positions the C2 atom of substrate precisely in relation to the N5 atom of FAD, enabling the catalytic reaction to take place. Besides, despite being a confined pocket, the subsite B can undergo conformational changes to varied extent to accommodate the small to moderate-size hydrophobic moiety of substrate. The moderate elasticity of the subsite B can explain why mLDHD exhibits tighter binding with lower $K_M$ (1.9–8.8 folds) and higher $k_{cat}/K_M$ (2.5–8.0 folds) for most of the substrates (D-2-hydroxybutyrate, D-2-hydroxyvalerate, D-2-hydroxyhexanoate, D-2-hydroxyisovalerate, D-2-hydroxyisocaproate, and D-2-hydroxy-3-methylvalerate) than for D-lactate as these substrates make more hydrophobic interactions with the protein. Nevertheless, as D-2-hydroxyoctanoate and D-2-hydroxy-3-phenylpropionate have larger hydrophobic moieties, the binding of these substrates pushes the conformational changes of the subsite B to the limit, which affects the $k_{cat}$ and/or $K_M$ values and thus decreases the catalytic efficiency ($k_{cat}/K_M$) by 2.5-fold and 4.3-fold, respectively.

On the other hand, glycolate makes no interactions with the subsite B despite containing the C1-carboxyl and C2-hydroxyl groups. Similarly, D-2-hydroxy-3-amino-propanoate, D-malate and D-2-hydroxyglutarate have a hydrophilic moiety attached to the C2 atom, which would not be able to bind favorably to the hydrophobic subsite B. As D-3-hydroxybutyrate contains a C3-hydroxyl group instead of a C2-hydroxyl group, it would lose the hydrophilic interactions of the C2-hydroxyl group with FAD, Mn$^{2+}$ and the protein. In addition, the C3-hydroxyl group would not be able to bind favorably to the hydrophobic subsite B. These results explain why mLDHD has no activity towards these substrate analogs.

To understand the molecular basis for why mLDHD has no activity for L-lactate and the L-isomers of other substrates, L-lactate was docked into the active site of mLDHD based on the mLDHD$^{FAD+Mn+D\text{-}LAC}$ structure (Supplementary Fig. 9). To avoid steric conflicts between L-lactate and FAD, the C2 atom of L-lactate is pushed away from the N5 atom of FAD (3.6 Å for L-lactate vs. 3.1 Å for D-lactate). Consequently, the C1-carboxyl and C2-hydroxyl groups of L-lactate are not in optimal positions to make hydrophilic interactions with Mn$^{2+}$ and the surrounding residues at the subsite A, and the C3 atom is not in proper position to make favorable hydrophobic interactions with the surrounding residues at the subsite B. This explains why mLDHD has high stereo-selectivity for D-lactate (and other D-2-hydroxyacids with hydrophobic moieties) against their L-isomers.

## Catalytic mechanism

Structural analyses of mLDHD in various forms reveal that residues His398, His405, and Glu442 are responsible for the Mn$^{2+}$ binding, and Arg347 and His443 for the substrate/product binding (Fig. 3b, d). Mutations of these residues to alanine all lead to complete abrogation of the activity, underscoring their functional roles in the catalytic reaction (Supplementary Table 1). Based on the previously proposed catalytic mechanism for other members of the VAO/PCMH flavoprotein family[46] and our structural and biochemical data, we can propose the catalytic mechanism for mLDHD to oxidize D-2-hydroxyacid into 2-ketoacid as follows (Fig. 3e). When the D-2-hydroxyacid substrate binds to the active site, the C1-carboxyl group is recognized by the sidechain of Arg347 via two salt-bridge bonds. In addition, the C1-carboxyl and the C2-hydroxyl groups form two coordination bonds with Mn$^{2+}$ to further stabilize the binding of substrate. The sidechain of His443 recognizes the C2-hydroxyl group and acts as a Lewis base to extract a proton from the hydroxyl group. Then, the hydride ion on the C2 atom is abstracted by the N5 atom of the flavin moiety of FAD, leading to formation of the C2-keto group and the flavin anion. Subsequently, the 2-ketoacid product is dissociated from the active site, and the reduced FAD is oxidized to FAD by an oxidant. In our in vitro activity assay, phenazine methosulfate (PMS) was used as the primary oxidant followed by 2,6-dichloroindophenol (DCIP). However, the primary oxidant under physiological conditions is unknown, and thus it remains unclear whether mLDHD would function as an oxidase when molecular oxygen is used as the oxidant or a dehydrogenase when other electron acceptor(s) are used as the oxidant in a coupled oxidation-reduction reaction. As the key residues involved in the metal ion and substrate binding and the catalytic reaction are highly conserved in other eukaryotic LDHDs (Supplementary Fig. 1), this catalytic mechanism is also applicable to other LDHDs.

## Functional roles of disease-associated mutations

D-lactic acidosis is considered as a complication of short-bowel syndrome for a long time[12,13,15]. Recently, it was reported that hLDHD mutations can also cause elevated concentrations of D-lactate and several other organic acids and consequently D-lactic acidosis[8,44,43]. However, the underlying molecular mechanism is unclear. Sequence alignment shows that hLDHD and mLDHD share high sequence similarity, and the four missense mutations identified in D-lactic acidosis

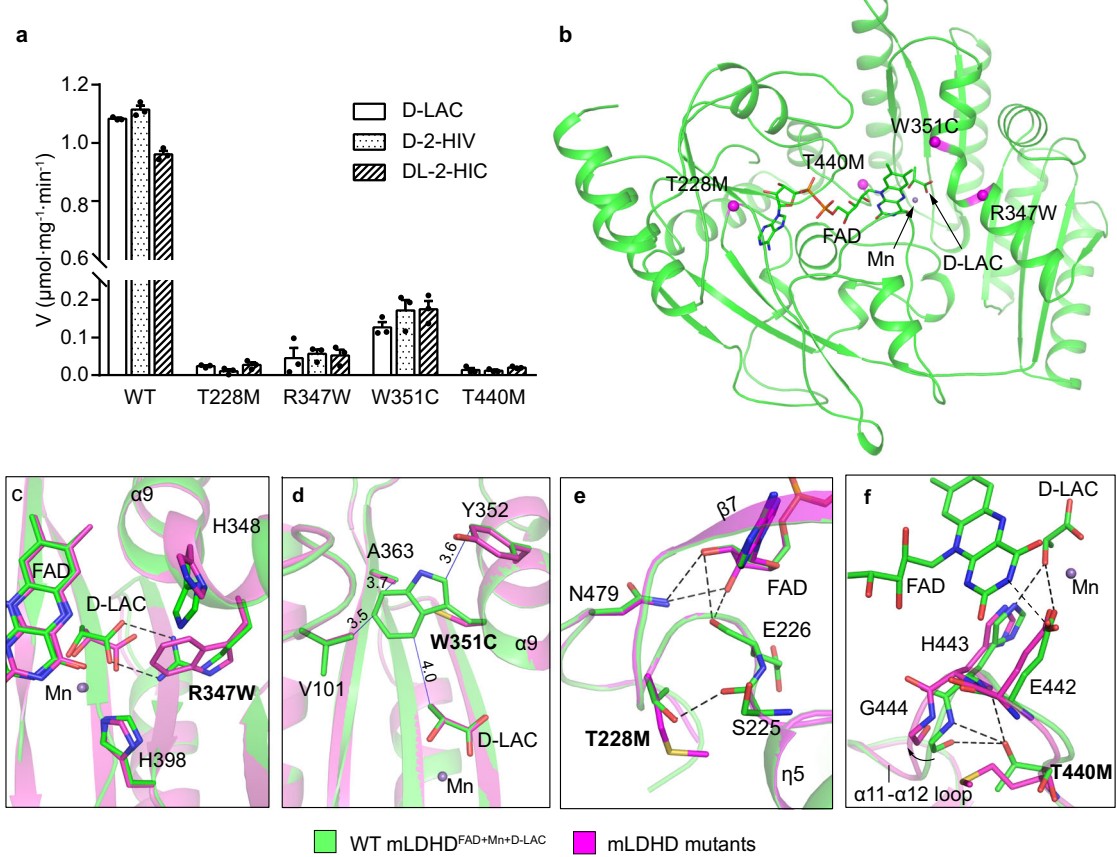

**Fig. 4 | Analysis of the functional roles of the disease-associated mutations of LDHD. a** Specific activity of WT mLDHD and the mLDHD$_{T228M}$, mLDHD$_{R347W}$, mLDHD$_{W351C}$ and mLDHD$_{T440M}$ mutants corresponding to the disease-associated hLDHD mutants using D-lactate (D-LAC), D-2-hydroxyisovalerate (D-2-HIV) or DL-2-hydroxyisocaproate (DL-2-HIC) as substrate. Data are presented as the mean ± SEM ($n = 3$ independent experiments). Source data are provided as a Source Data file. **b** Locations of the mutations of mLDHD corresponding to the disease-associated mutations of hLDHD. The mutations are indicated with magenta spheres. **c–f** Zoom-in views showing the impacts of the mutations on the structure of mLDHD and its interactions with the surrounding residues, the substrate and/or FAD: R347W (**c**), W351C (**d**), T228M (**e**), and T440M (**f**). The WT mLDHD$^{FAD+Mn+D-LAC}$ structure is colored in green. The predicted structure models of the mLDHD mutants are shown in magenta. The hydrogen bonds in the mLDHD$^{FAD+Mn+D-LAC}$ structure are shown as dashed lines, and the hydrophobic interactions are shown as blue solid lines and indicated with distances.

patients are strictly conserved (Supplementary Fig. 1). The T251M, R370W, W374C, and T463M mutations of hLDHD correspond to T228M, R347W, W351C, and T440M of mLDHD, respectively. To understand the molecular basis of these missense mutations of hLDHD in the pathogenesis of D-lactic acidosis, we performed biochemical and structural analyses of the mLDHD mutants.

Biochemical data show that all the mLDHD mutants containing the above mutations can be expressed and purified (Supplementary Fig. 10a). Compared to WT mLDHD, the melting temperatures (T$_m$) of the mutants are decreased by 10-15 °C, indicating that these mutations impair the thermostability of mLDHD (Supplementary Fig. 10b). In addition, size-exclusion chromatography analysis shows that compared to WT mLDHD, the R347W and W351C mutants retain the FAD characteristic absorption peak, indicative of FAD binding albeit with lower occupancies; in contrast, the T228M and T440M mutants do not exhibit the FAD characteristic absorption peak, suggesting the absence of FAD binding (Supplementary Fig. 10c–f). Moreover, the activity assay shows that all the mutants lose about 90% of the activity towards D-lactate, D-2-hydroxyisovalerate and DL-2-hydroxyisocaproate (Fig. 4a and Supplementary Table 1).

Comparison of the Alphafold2 predicted hLDHD structure and the mLDHD$^{FAD+Mn+D-LAC}$ structure reveals an RMSD of 0.69 Å over 398 aligned Cα atoms (Supplementary Fig. 11). Compared to mLDHD, hLDHD contains an extra insertion (residues 210–232) which assumes a flexible

loop between β6 and η4 of mLDHD on the protein surface. This insertion is positioned far away from the FAD-binding site and the substrate-binding site, and thus is unlikely to affect the binding of FAD and/or substrate as well as the catalytic reaction (Supplementary Fig. 11). To examine the functional roles of the disease-associated mutations of hLDHD in the pathogenesis of D-lactic acidosis, we mapped the corresponding mutations on the mLDHD$^{FAD+Mn+D-LAC}$ structure and examined their effects on the structure and function of mLDHD (Fig. 4b).

In the mLDHD$^{FAD+Mn+D-LAC}$ structure, Arg347 and Trp351 form part of the substrate-binding site (Fig. 4b). Arg347 is located in the α9 helix and is a key residue at the substrate-binding subsite A to recognize and bind the C1-carboxyl group of D-lactate. Modeling study shows that the R347W mutation causes steric conflicts with the substrate and thus disrupts the substrate binding (Fig. 4c). Trp351 is also located in the α9 helix and forms part of the substrate-binding subsite B. It makes several hydrophobic interactions with the C3-moiety of D-lactate and several surrounding residues, including Val101, Ala363, and Tyr352 at the subsite B (Fig. 4d). The W351C mutation disrupts these hydrophobic interactions, leading to destabilization of the conformation of the subsite B and thus impairment of the substrate binding. Nevertheless, the R347W and W351C mutations do not directly interfere with the FAD binding, which is consistent with the biochemical data showing that the two mutants can bind FAD albeit with lower occupancies (Supplementary Fig. 10c, d).

Thr228 and Thr440 of mLDHD are both located near the FAD-binding site (Fig. 4b). Thr228 of mLDHD is located in the η4-β7 loop of the FAD-binding subdomain B. Although Thr228 makes no direct interaction with FAD, it forms a hydrogen bond with the mainchain of Ser225 to stabilize the conformation of the η4-β7 loop (Fig. 4e). As Glu226 on this loop has direct interactions with FAD, the T228M mutation might destabilize the conformation of the η4-β7 loop and hence affect the FAD binding. In addition, as Thr228 is positioned in the close vicinity of Asn479 on the C-terminal domain, the T228M mutation might also affect the interactions of the C-terminal domain with FAD. Thr440 of mLDHD is located in the α11-α12 loop connecting the substrate-binding domain and the C-terminal domain. The side-chain of Thr440 forms hydrogen bonds with the mainchains of His443 and Gly444 to stabilize the conformation of the loop. His443 forms a hydrogen bond with the C2-hydroxyl group of D-lactate, and Glu442 makes direct interactions with both FAD and substrate, and coordinates the metal ion (Fig. 4f). The T440M mutation causes steric clashes with the backbones of His443 and Gly444, leading to distortion of the α11-α12 loop and displacements of His443 and Glu442, which subsequently interfere with the binding of FAD, substrate and metal ion. These results suggest that the T228M and T440M mutants impair the FAD binding, which is consistent with the biochemical data showing that these two mutants lose the FAD-binding ability (Supplementary Fig. 10e, f).

Taken together, the structural and biochemical analysis results indicate that the disease-associated mutations either disrupt the substrate binding (R347W and W351C) or the FAD binding (T228M and T440M), leading to abolishment of the enzymatic activity. Thus, all of these mutations are pathogenic in the development of D-lactic acidosis.

## Discussion

Previously, LDHD was reported to catalyze the conversion of D-lactate to pyruvate. In this study, we performed systematic biochemical and structural studies of mLDHD towards a variety of 2-hydroxyacids. Our biochemical data demonstrate that mLDHD is an $Mn^{2+}$-dependent general dehydrogenase for D-2-hydroxyacids with hydrophobic moieties at the C2 position. Our structural data identify the key residues involved in the substrate recognition and catalytic reaction and reveal the structural basis for the high substrate specificity of mLDHD. Moreover, our data provide the molecular basis for the functional roles of hLDHD mutations found in patients with D-lactic acidosis.

LDHD is evolutionarily distinct from the L-lactate dehydrogenases and NAD-dependent D-lactate dehydrogenases. The search of homologous proteins sharing similar structures with mLDHD in the Protein Data Bank (PDB) via DALI server[48] identified two best candidates: *A. woodii* LctD (PDB: 7QH2)[49] with a Z-score of 47.2 and human D2HGDH (PDB: 6LPN)[40] with a Z-score of 44.9. Although *A. woodii* LctD and human D2HGDH share only a sequence identity of 29% and 27% with mLDHD, respectively, they adopt similar overall structures consisting of an FAD-binding domain, a substrate-binding domain and a C-terminal domain. Superposition of mLDHD with *A. woodii* LctD and human D2HGDH reveals an RMSD of 1.7 Å (for 386 aligned Cα atoms) and 1.6 Å (for 338 aligned Cα atoms), respectively. *A. woodii* LctD is a homolog of mLDHD, and thus it also contains a highly hydrophobic substrate-binding subsite B[49] and exhibits high substrate specificity for D-lactate against L-lactate[50].

Evolutionarily, LDHD, D2HGDG, and GlcD are classified into the 2-hydroxyacid dehydrogenase subfamily of the VAO/PCMH flavoprotein family[39]. Although these enzymes share similar structural features, they display distinct substrate specificity. GlcD is only found in bacteria and can oxidize both glycolate and D-lactate with a 10-fold higher binding affinity for glycolate than D-lactate[51]. Human D2HGDH has high activity for D-2-hydroxyglutarate and D-malate[40]. Mouse LDHD has a broad specificity for D-2-hydroxyacids containing hydrophobic

moieties at the C2 atom. The differed substrate specificities could be explained by their distinct structural features of the substrate-binding sites. In the crystal structures of mLDHD and human D2HGDH, and the Alphafold2 predicted structure of *E. coli* GlcD, the residues composing the substrate-binding subsite A are highly conserved, which interact with the C1-carboxyl and C2-hydroxyl groups of substrates in a similar manner (Supplementary Fig. 12). However, there are significant differences in the substrate-binding subsite B of LDHD, D2HGDH and GlcD. The subsite B of mLDHD is a large, hydrophobic pocket consisting of Trp351, Ile407, Val101 and Ser365 with considerable flexibility to accommodate a variety of hydrophobic moieties of small to medium size (Fig. 3a). On the other hand, the subsite B of D2HGDH is a deep, positively charged pocket framed by Thr390, Asn443 and Lys401 (Supplementary Fig. 12a, b)[40]. Thus, the subsite B of D2HGDH recognizes the C5-carboxyl group of D-2-hydroxyglutarate (or C4-carboxyl group of D-malate) with hydrophilic interactions, explaining D2HGDH's high substrate specificity for D-2-hydroxyglutarate and D-malate (Supplementary Fig. 12c)[40]. Compared to LDHD, the subsite B of GlcD is a shallow, hydrophobic pocket (Supplementary Fig. 12d). Notably, Ala363 in mLDHD is replaced by Tyr351 in GlcD. The larger sidechain of Tyr351 causes a steric clash with Phe341 (corresponding to Trp351 in LDHD), which makes the sidechain of Phe341 shift towards the subsite B (Supplementary Fig. 12e, f). Docking study shows that glycolate could be docked into the substrate-binding site of GlcD, while the other substrate analogs with longer or larger moieties at the C2 atom would have structural clashes with Phe341, explaining GlcD's high substrate specificity for glycolate.

Previously, the function of LDHD has been linked with D-lactate metabolism. In humans, D-lactate can be derived from food, metabolism of intestinal bacteria, or endogenous glyoxalase pathway. The miniscule amounts of D-lactate can be converted into pyruvate by LDHD and the product is an important metabolite in the tricarboxylic acid cycle[52]. Abnormal accumulation of D-lactate causes D-lactic acidosis. Interestingly, we demonstrate that LDHD can effectively catalyze the conversion of several branched-chain D-2-hydroxyacids such as D-2-hydroxyisovalerate, D-2-hydroxyisocaproate and D-2-hydroxy-3-methylvalerate, into 2-ketoisovalerate, 2-ketoisocaproate and 2-keto-3-methylvalerate, respectively. These products are collectively referred to as branched-chain ketoacids (BCKAs), which are also catabolic products of branched-chain amino acids (BCAAs). Previously, it is well established that BCKAs are produced from transamination of BCAAs catalyzed by branched-chain amino acid aminotransferase (BCAT), and are decarboxylated by branched-chain ketoacid dehydrogenase (BCKDH). Thus, BCKAs participate in the regulation of various metabolic pathways[53]. For example, 3-hydroxyisobutyrate, the metabolic product of 2-ketoisovalerate, is found to be a downstream signaling molecule of BCAA catabolism in muscle[54]. Elevated BCAA catabolic flux can cause elevated secretion of 3-hydroxyisobutyrate, resulting in excess intake of fatty acids, accumulation of lipotoxicity, and impairment of insulin signaling[54]. On the other hand, BCKAs can allosterically inhibit BCKDH kinase (BCKDK) and thus alleviate the suppression of BCKDH activity by BCKDK, which in turn regulates the rate of BCAA metabolic pathway[53]. In this work, we demonstrate that besides BCAA catabolism, BCKAs can also be generated by oxidation of branched-chain D-2-hydroxyacids (D-2-hydroxyisovalerate, D-2-hydroxyisocaproate and D-2-hydroxy-3-methylvalerate) catalyzed by LDHD. Moreover, abnormal accumulation of D-2-hydroxyisovalerate, D-2-hydroxyisocaproate and 2-hydroxy-3-methylvalerate are found in patients with defective LDHD containing loss-of-function mutations, whereas these organic acids are almost undetectable under normal physiological conditions[8,44]. Our data and the previous findings together suggest that LDHD can convert D-2-hydroxyisovalerate, D-2-hydroxyisocaproate and D-2-hydroxy-3-methylvalerate to the corresponding BCKAs to keep these branched-chain D-2-hydroxyacids at trace levels to avoid harmful pathogenic effects under normal

physiological conditions. In addition, the yielded BCKA products may be involved in the regulation of BCAA metabolic pathway. Thus, LDHD may play an important role not only in the metabolism of branched-chain D-2-hydroxyacids but also the metabolism of BCAAs.

In this study, we also identified a series of linear-chain D-2-hydroxyacids as the substrates of LDHD, including D-2-hydroxybutyrate, D-2-hydroxyvalerate, D-2-hydroxyhexanoate and D-2-hydroxyoctanoate. In the human body, these substances are either derived from food or produced by intestinal bacteria, but no endogenous source has been reported. Previously, human fatty acid 2-hydroxylase (FA2H) has been reported to catalyze the stereospecific hydroxylation of free fatty acids at the C2 atom to produce D-2-hydroxy fatty acids[55–57]. Although so far there is no biochemical data showing that FA2H is capable of producing the LDHD substrates identified in this work, it is worthy to explore the possibility whether the biochemical reaction catalyzed by FA2H could be the source of LDHD substrates, and FA2H and LDHD could function together to play some roles in the metabolism of fatty acids.

LDHD deficiency is caused by loss-of-function mutations[8,43,44]. Clinically, LDHD deficiency is marked by elevated levels of D-lactate in urine and plasma and manifested by neurological symptoms, hyperuricemia and gout[8,44]. Currently, clinical diagnosis of LDHD deficiency is focused on urine and plasma D-lactate levels. However, other diseases such as short bowel syndrome may also lead to accumulation of D-lactate, which can be confused with that caused by LDHD mutations. So far, the main treatments for D-lactic acidosis due to short bowel syndrome include antibiotic therapy, diet control, bicarbonate therapy and hemodialysis through strict control of D-lactate production[10]. However, there is no treatment for symptoms specifically caused by LDHD deficiency. In addition, LDHD mutations also lead to elevated levels of other 2-hydroxyacids such as D-2-hydroxyisovalerate, D-2-hydroxyisocaproate and 2-hydroxy-3-methylvalerate, which may have impacts on the onset and symptoms in LDHD deficiency. Therefore, therapies aiming at control of D-lactate production may not be an effective way for LDHD deficiency. This study largely expands the substrates of LDHD, showing that LDHD can directly regulate the levels of D-2-hydroxyisovalerate, D-2-hydroxyisocaproate and D-2-hydroxy-3-methylvalerate. These substrates can serve as potential biomarkers of LDHD deficiency, providing differential diagnostic criteria for D-lactic acidosis patients with LDHD mutations or with short bowel syndrome. In addition to D-lactic acidosis, down-regulation of LDHD expression is related to poor prognosis of many tumors such as cholangiocarcinoma and renal cell carcinoma[58,59]. Besides, elevated LDHD expression is found in prostate cancer cells, which exhibit higher mitochondrial D-lactate metabolism than normal cells[60]. These findings suggest possible functional role of LDHD in both physiological processes and cancer metabolism. The potentials of LDHD and its substrates/products as diagnostic markers and therapeutic targets deserve further investigation. The physio-pathological role of these substrates and the regulatory role of LDHD also deserve further investigation.

## Methods

### Cloning, expression, and purification of mLDHD
The mouse *Ldhd* gene was amplified by PCR from the cDNA library of mouse liver cells. The mouse *Ldhd* gene fragment encoding residues 22–484 was cloned into the pET28 vector (Novagen) attached with a Strep tag at the N-terminus. The N-terminal 1–21 residues of mLDHD are suggested to be a consensus mitochondrial targeting sequence and thus were removed. The truncated protein is more stable than the full-length protein. The plasmid was transformed into *E. coli* BL21 (DE3) CodonPlus strain (Weidi Bio). Protein over-expression was induced with 0.2 mM IPTG at 16 °C overnight after the transformed cells grew in LB medium supplemented with 0.05 mg/ml kanamycin at 37 °C and OD$_{600}$ reached 0.8. The *E. coli* cells were collected by centrifugation

and lysed in a lysis buffer (50 mM Tris-HCl, pH 7.4, 200 mM NaCl, 5% glycerol, 1 mM EDTA, 14 mM 2-mercaptoethanol, and 1 mM PMSF) using a sonicator. The recombinant protein was purified by affinity chromatography using Streptactin Beads 4FF (Smart-Lifesciences) followed by gel filtration chromatography using a Superdex 200 10/300 column (GE Healthcare). The elution curve was tracked using Unicorn 7 software integrated with AKTA chromatography system (Cytiva). The purified protein was stored in a storage buffer (50 mM Tris-HCl, pH 7.4, 200 mM NaCl, and 1 mM EDTA) for structural and biochemical studies. 1 mM EDTA was added in the lysis buffer and the storage buffer to remove potential unknown divalent metal ion(s) bound to mLDHD from expression and purification. The mLDHD mutants containing point mutations were generated using 2 x Phanta Flash Master Mix (Vazyme Biotech Co., Ltd) and confirmed by DNA sequencing. Expression and purification of the mutants were the same as the wild-type protein.

### Enzymatic activity assay
D-lactate (71716), pyruvate (P2256), D-2-hydroxyisovalerate (55452), DL-2-hydroxyisocaproate (219819), DL-2-hydroxy-3-methylvalerate (80529), L-lactate (L7022), D-2-hydroxyglutarate (H8378), 2-ketoisovalerate (198994), and 2-ketoisocaproate (K0629) were purchased from Merck. Glycolate (1199045), D-2-hydroxybutyrate (1178491), DL-2-hydroxyvalerate (1132981), DL-2-hydroxyhexanoate (1058880), DL-2-hydroxyoctanoate (1037878), D-2-hydroxy-3-phenyl-propionate (1041025), D-malate (1038701), and DL-2-hydroxy-3-amino-propanoate (1036104) were purchased from Shanghai Haohong Scientific Co. Ltd.

The enzymatic activity of mLDHD was determined using a PMS-DCIP coupled assay described previously[40]. Specifically, the reduced FAD in the catalytic reaction was oxidized by phenazine methosulfate (PMS, Sangon Biotech, A610361) and subsequently, the reduced PMS was oxidized by 2,6-dichloroindophenol (DCIP, Sangon Biotech, A600396). The reduction of DCIP was measured spectrophotometrically at 600 nm using a Synergy Neo2 Hybrid Multi-Mode Reader (BioTek Instruments).

To determine the specific activity, the standard reaction solution (200 µL) consisting of 50 mM Tris-HCl (pH 7.4), 2 µg enzyme, 50 µM MnCl$_2$, 200 µM PMS, 100 µM DCIP, and 1 mM substrate (D-lactate, D-2-hydroxybutyrate, DL-2-hydroxyvalerate, DL-2-hydroxyhexanoate, DL-2-hydroxyoctanoate, D-2-hydroxyisovalerate, DL-2-hydroxyisocaproate, DL-2-hydroxy-3-methylvalerate, and D-2-hydroxy-3-phenylpropionate) or analog (L-lactate, glycolate, DL-2-hydroxy-3-amino-propanoate, D-malate, D-2-hydroxyglutarate, and DL-3-hydroxybutyrate) incubated at 37 °C using 96-well plate (Corning). The catalytic reaction was initiated by addition of the substrate or analog. For the racemic substrates, the substrate concentration was doubled and the specific activity was calculated based on the corrected concentration of D-isomer (1/2 of the concentration of racemic substrate).

To determine the saturation curve, the standard reaction solution was provided with varied concentrations (0-2 mM) of substrate. The activity is defined as the micromoles of DCIP reduced per min per milligram of enzyme (µmol•min$^{-1}$•mg$^{-1}$). To analyze the effect of different divalent metal ions on the activity, the specific activity of mLDHD was measured in the absence of any metal ions or the presence of eight commonly used divalent metal ions (1 mM Mn$^{2+}$, Mg$^{2+}$, Ni$^{2+}$, Co$^{2+}$, Zn$^{2+}$, Cd$^{2+}$, Cu$^{2+}$, and Ca$^{2+}$). To determine the optimum Mn$^{2+}$ concentration of the catalytic reaction, the Mn$^{2+}$ concentration was varied in the range of 5-1000 µM. To determine the optimum pH of the catalytic reaction, the Tris-HCl (50 mM, pH 7.4) in the standard reaction solution was substituted with the phosphate buffer (66 mM, pH 5.6–8.6). The kinetic parameters ($V_{max}$, $K_M$, and $k_{cat}$) were derived by fitting the kinetic data into the Michaelis-Menten equation "$V = V_{max}*[S]/(K_M + [S])$" using program Graphpad Prism 7.0 (Graphpad Software).

The concentration of the mLDHD protein was determined using the coextinction coefficient of protein at A280 (42,565 $M^{-1}cm^{-1}$). The concentration of bound FAD in the mLDHD protein was determined using a standard curve measured at A450. The molar ratio of FAD:mLDHD in the WT mLDHD protein was determined to be $0.73 \pm 0.04$ (mean $\pm$ SEM of three independent measurements). The activity and kinetic parameters of mLDHD were corrected according to the concentration of active enzyme in the reaction solution calculated based on the FAD occupancy.

All experiments were performed in triplicates using distinct samples, and the values were the means of the triplicate measurements with the standard errors. The values of kinetic parameters were calculated from the saturation curve of the averages of the triplicate measurements with the fitting errors. $P$ values were calculated with two-sided Student's $t$-test.

### Crystallization, data collection, and structure determination

Equal volume (1 μl) of the mLDHD solution (8 mg/ml) and the reservoir solution were mixed, and crystallization was set up at 16 °C using the hanging drop vapor diffusion method. Crystals of wild-type (WT) mLDHD and mLDHD$_{H405A}$ bound with FAD were grown from drops containing the reservoir solution of 4.0 M sodium formate. Crystals of WT mLDHD in complex with FAD, $Mn^{2+}$, and D-lactate were obtained by soaking the FAD-bound WT mLDHD crystals in the crystallization solution supplemented with 50 mM $MnCl_2$ and 50 mM D-lactate for 15−30 min. Crystals of mLDHD$_{H405A}$ in complexes with FAD and substrates (D-lactate, D-2-hydroxybutyrate, D-2-hydroxyvalerate, D-2-hydroxyhexanoate, D-2-hydroxyoctanoate, D-2-hydroxyisovalerate, D-2-hydroxyisocaproate, and D-2-hydroxy-3-methylvalerate) were obtained by soaking the FAD-bound mLDHD$_{H405A}$ crystals in the crystallization solution supplemented with 50 mM $MnCl_2$ and 20-50 mM D-lactate, D-2-hydroxybutyrate, DL-2-hydroxyvalerate, DL-2-hydroxyhexanoate, DL-2-hydroxyoctanoate, D-2-hydroxyisovalerate, DL-2-hydroxyisocaproate, and DL-2-hydroxy-3-methylvalerate for 30−60 min, respectively. Crystals of WT mLDHD in complexes with FAD, $Mn^{2+}$ and products (pyruvate, 2-ketobutyrate, 2-ketovalerate, 2-ketohexanoate, 2-ketoisovalerate, 2-ketoisocaproate, and 2-keto-3-methylvalerate) were obtained by soaking the FAD-bound WT mLDHD crystals in the crystallization solution supplemented with 50 mM $MnCl_2$ and 20−50 mM product for 30−60 min or substrate for 1−2 h. Soaking of the FAD-bound mLDHD$_{H405A}$ crystals in crystallization solution supplemented with DL-2-hydroxy-3-phenylpropionate did not yield the ligand-bound structure probably due to the weak binding of DL-2-hydroxy-3-phenylpropionate caused by its large-size hydrophobic moiety. Soaking of the FAD-bound WT mLDHD crystals in crystallization solution supplemented with L-lactate, glycolate, D-malate, D-2-hydroxyglutarate, D-2-hydroxy-3-amino-propanoate, or D-3-hydroxybutyrate did not yield the ligand-bound structure, suggesting that these ligands cannot bind to mLDHD.

Prior to diffraction data collection, the crystals were cryoprotected using the reservoir solution plus 20% glycerol and then flash-cooled in liquid $N_2$. Diffraction data were collected at 100 K at beamlines of Shanghai Synchrotron Radiation Facility and processed by autoPROC[61], HKL2000 (v718)[62] and XDS[63]. The FAD-bound WT mLDHD structure was solved by the molecular replacement (MR) method implemented in Phenix (1.18.2)[64] using the Alphafold2 predicted mLDHD structure (Q7TNG8) as the search model. The structures of WT mLDHD in complexes with D-lactate and various products and the mLDHD$_{H405A}$ mutant in FAD-bound form and in complexes with various substrates were solved by the MR method using the FAD-bound WT mLDHD structure as the search model. Manual model building was performed using Coot (0.8.2)[65] and structure refinement was performed using Phenix (1.18.2)[64]. Statistics of diffraction data collection, structure refinement and the quality of final structure models are summarized in Supplementary Data 1.

The structure models of the mLDHD mutants containing T228M, R347W, W351C, and T440M mutations were built using the crystal structure of mLDHD$^{FAD+Mn+D-LAC}$ as template by YASARA suit (17.8.15)[66,67]. Energy minimization was performed for all mutant structures in vacuo to fix structural bumps after mutation[66,67]. The structure models of *E. coli* GlcD in complex with glycolate and mLDHD in complex with L-lactate were built based on the Alphafold2 predicted structure of GlcD (P0AEP9) and the crystal structure of mLDHD$^{FAD+Mn+D-LAC}$, respectively. Docking experiments were performed using YASARA (17.8.15)[66,67]. Structural analysis and structural figure preparation were carried out using PyMOL (2.4.1)[68].

### Protein thermostability analysis

The protein solution (wild-type and mLDHD mutants) was diluted with the storage buffer to a concentration of 1 mg/ml. The sample (50 μl) was loaded with high precision quartz glass capillary (Shanghai Yuanyi Biotechnology). Protein thermostability was determined using a Prometheus NT.48 instrument (NanoTemper Technologies). The temperature range was 25−95 °C with a rise of 1 °C per min. Fluorescence intensities were measured at 330 nm and 350 nm. Data were analyzed using PR.ThermControl v2.3.1 (NanoTemper Technologies). All experiments were performed in triplicates using distinct samples, and the $T_m$ values were the averages of the triplicate measurements with the standard errors.

### Reporting summary

Further information on research design is available in the Nature Portfolio Reporting Summary linked to this article.

## Data availability

Atomic coordinates and structure factors of the following structures have been deposited with the Protein Data Bank (PDB): mLDHD in complex with FAD (PDB 8JDC), mLDHD$_{H405A}$ in complex with FAD (PDB 8JDD), mLDHD in complex with FAD, $Mn^{2+}$ and D-lactate (PDB 8JDE), mLDHD$_{H405A}$ in complex with FAD and D-lactate (PDB 8JDF), mLDHD$_{H405A}$ in complex with FAD and D-2-hydroxybutyrate (PDB 8JDG), mLDHD$_{H405A}$ in complex with FAD and D-2-hydroxyvalerate (PDB 8JDN), mLDHD$_{H405A}$ in complex with FAD and D-2-hydroxyhexanoate (PDB 8JDO), mLDHD$_{H405A}$ in complex with FAD and D-2-hydroxyoctanoate (PDB 8JDB), mLDHD$_{H405A}$ in complex with FAD and D-2-hydroxyisovalerate (PDB 8JDP), mLDHD$_{H405A}$ in complex with FAD and D-2-hydroxyisocaproate (PDB 8JDQ), mLDHD$_{H405A}$ in complex with FAD and D-2-hydroxy-3-methylvalerate (PDB 8JDR), mLDHD in complex with FAD, $Mn^{2+}$ and pyruvate (PDB 8JDS), mLDHD in complex with FAD, $Mn^{2+}$ and 2-ketobutyrate (PDB 8JDT), mLDHD in complex with FAD, $Mn^{2+}$ and 2-ketovalerate (PDB 8JDU), mLDHD in complex with FAD, $Mn^{2+}$ and 2-ketohexanoate (PDB 8JDV), mLDHD in complex with FAD, $Mn^{2+}$ and 2-ketoisovalerate (PDB 8JDX), mLDHD in complex with FAD, $Mn^{2+}$ and 2-ketoisocaproate (PDB 8JDY), and mLDHD in complex with FAD, $Mn^{2+}$ and 2-keto-3-methylvalerate (PDB 8JDZ). All other data are contained within the manuscript and the supplementary information file. Source data are provided with this paper.

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

## Acknowledgements

We thank the staff members at BL02U1 of Shanghai Synchrotron Radiation Facility (SSRF) and BL18U and BL19U1 of National Facility for Protein Science in Shanghai (NFPSS) for technical assistance in diffraction data collection, and other members of our group for valuable discussion. This work was supported by grants from the Chinese Academy of Sciences (grant number: XDB37030305; J.D.) and the National Natural Science Foundation of China (grant number: 31870723; J.D.).

## Author contributions

S.J. carried out the biochemical and structural studies, and participated in the data analyses and paper writing. X.C. carried out the data analyses and figure preparation and participated in paper writing. J.Y. participated in the initial structural studies. J.D. conceived the study, participated in the experimental design and data analyses, and wrote the manuscript.

## Competing interests

The authors declare no competing interests.
