## [Peer Review File · Nature Communications]

REVIEWER COMMENTS

Reviewer #1 (Remarks to the Author):

The manuscript reports in the biochemical and structural investigation of mouse D-lactate dehydrogenase, a recently discovered mammalian (including human) flavoenzyme. The paper unveils two main features: the enzyme performs best using manganese as metal cofactor and is active on several 2-hydroxy acid substrates, all sharing an aliphatic/hydrophobic side chain (i.e. the hydroxy equivalents of the branched-chain amino acids).

The work is relevant because it demonstrates that D-hydroxy acids can be a source of branch-chain alpha-keto acids and the three-dimensional structure nicely rationalizes the basis for the substrate selectivity of the enzyme. An additional point of interest is that the structural and biochemical analysis explains the pathogenicity of the mutations causing the enzyme deficiency, a genetic disease. The mutations negatively impact on the enzyme stability and affect residues that are directly or indirectly involved in catalysis. The other points made by the manuscript are mostly confirmatory. The catalytic mechanism is confirmed to involve a hydride transfer step whereas the FAD-binding site is confirmed to be similar to that of other flavoenzymes.

Methodologically, the work is sound and the crystallographic investigation is of the finest quality.

Comments

-The manuscript is long and occasionally repetitive. I would suggest to shorten it to make it more focused on what is really new, and possibly unanticipated. For instance, the text extensively discusses the FAD binding site that hardly reveals anything new. The basis of the stereo-selectivity is the same as discussed before for other oxidases and dehydrogenases. The Discussion repeats the points that have been analysed and discussed in the Results. Better to focus the Discussion on what is new and has been learned from this work.

-The overall enzyme structure should be better compared to that of other enzymes. What are the most similar enzyme structures present in the PDB? What are the differences?

-The manganese dependency is intriguing. Do our cells contain enough manganese to saturate the enzyme that binds the metal with a 50 μM K_d ?

-Please avoid an excessive use of abbreviations. I found the manuscript often hard to read because of this. I do not see any reason for not using the standard nomenclature for the substrates and products.

-Did the authors check if the enzyme works also with cytochrome c or other non-oxygen electron acceptors such as DCPIP? Can it work as a dehydrogenase?

Reviewer #2 (Remarks to the Author):

In this manuscript the authors perform steady-state kinetic and structural characterization of mouse lactate dehydrogenase (mLDHD) and disease associated variants, showing it has specificity to D-lactate and a broad range of hydrophobic D-2-hydroxyacids. The X-ray crystal structures were determined with FAD, Mn²⁺ and a series of substrates or products. The most interesting aspect of the work is that the enzyme may play an important role in the metabolism of these hydroxyacids and the part that disease-associated mutations of mLDHD may play in the pathogenesis of D-lactic acidosis. Indeed, knowledge of the potential substrates and products may be useful diagnostically. However, there are some concerns about some of the kinetic analyses and interpretations of the authors. It is not clear why the crystalline enzyme has activity in the unmetallated form (where they have treated with EDTA) and this raises the question of whether the divalent cation is a cofactor or an activator. Is the enzyme simply contaminated with Mn²⁺ leading to the observed activity or is Mn²⁺ unnecessary? The authors kinetics do not take into account the occupancy of FAD so it is uncertain if the k_{cat} values are correct; the ratio of enzyme to FAD cofactor is needed to know the concentration of active enzyme. The manuscript could use another pass for grammar and usage. The authors may wish to consider the following points (page numbers from pdf for review):

1) The authors state (pg 6) "Later structure determination of mLDHD in apo form confirms that there is an FAD but no metal ion bound "The word apo here is incorrectly used as that describes cofactor-free enzyme and FAD is bound. This should be described as metal free or Mn²⁺ free here and throughout the manuscript.

2) the rates and k_{cat} given do not support the claim of a high catalytic activity- the k_{cat}/K_m is $6 \times 10^4 \text{ M}^{-1} \text{ s}^{-1}$ which falls below the numbers expected for high catalytic activity of 10^7 - $10^9 \text{ M}^{-1} \text{ s}^{-1}$ (diffusion controlled) and below the average of most physiological enzymes $10^5 \text{ M}^{-1} \text{ s}^{-1}$ [see Biochemistry (2011) ,50, pp 4402–4410]. The authors should address this activity with comparisons to the literature to insure that it fits with the physiological role hypothesized for this enzyme

3) Page: 7- It would be much easier to follow the arguments about substrate specificity if the ChemDraw Figures in S4 had the kinetic parameters (velocity) from 1D beside each structure

4) Page: 8- The authors state "mLDHD exhibits comparable activities and kinetic parameters compared to the pure D isomer substrates with similar chemical structures, indicating that the presence of L isomers has no effect on the activity of mLDHD towards the D-isomers and implying that the L-isomers cannot bind to mLDHD" I assume that when the authors state this that they are comparing the corrected concentration for the racemic mixtures (2 fold lower than pure isomers). This should be clarified.

5) Page: 9- "As mLDHD has high activity towards the substrates, soaking of the apo WT mLDHD crystals in crystallization solution supplemented with the substrates often yielded the product-bound or mixed substrate/product-bound structures" This is a confusing statement compared to results in Figure S3, which I think shows no activity in the absence of Mn^{2+} . If the enzyme is active in the absence of Mn^{2+} then the ion is not a cofactor but an activator. Please clarify throughout the manuscript. Are other enzymes in the enzyme family or subfamily (ie. LDHD, D2HGDG and GlcD) dependent on divalent cation?

6) Can the authors comment if the poorly defined surface residues are still included in the structure?

7) The ratio of FAD bound to protein should be quantified by measurement of FAD absorption (OD 450) and protein assay. The k_{cat} values may need to be corrected in cases where FAD occupancy is low.

8) The authors state- "Although the DL-isomer mixtures of some substrates (DL-2-HV, DL-2-HH, DL-2-HO, DL-2-HIC, and DL-2-HMV) were used in the crystallization experiments, only the D-isomers are found to bind to the active site, further confirming that the L-isomers cannot bind to mLDHD." This is not exactly true depending on the relative K_d s of the two isomers, one may simply be competing out the other. Especially as a very low occupancy (<5%) the L-isomer may not be detected in the X-ray structure.

9) The authors present the proposed mechanism of the enzyme in Figure 3F but has this mechanism been proposed or differ from the mechanisms proposed for other subfamily members? This should be in the Discussion.

Minor grammatical errors/typos, missing labels

Page: 3- Accumulation of excess L- and D-LAC in human body can cause a disease called lactic acidosis" should read "Accumulation of excess L- and D-LAC in the human body ..."

Page: 4- "other organic acids in urea and plasm" I think this should read "other organic acids in urea and plasma." This also occurs on other pages- please fix all occurrences of plasm to plasma

Page: 5- "...only insoluble inclusion body" should read "...only insoluble inclusion bodies"

Page 6 - which is agreement with the weak alkaline physiological environment in vivo" should read "which is in agreement with the weak alkaline ..."

Page: 10- "which comprises of two subdomains" should read "which is comprised of two..."

Page: 11- The substrate-binding site is located in opposite to the si face" should read "...located opposite to the ..."

Page: 12- In Fig 3C the length of the Mn-H405 bond is missing

Page: 13- In Fig 2E I can't tell if 2.3 corresponds to the bond to His405 or lactate- one is missing.

Page 13- "the substrate binding and posit the C2 atom of the substrate in precise position should read "the substrate binding and position the C2 atom of the substrate precisely..."

Page 17- The authors state "Residues Thr251 and Thr440 of mLDHD are both located near the FAD-binding site (Fig. 4B)." Thr251 is not shown in the figure.

Page: 20- "In addition, the yielding BCKA products might be involved in the regulation of the BCAA metabolic pathway"- should read "In addition, the yielded BCKA products might be involved in the regulation of the BCAA metabolic pathway"

Point-by-Point Responses to Reviewers' Comments

Reviewer #1

Overall comments:

The manuscript reports in the biochemical and structural investigation of mouse D-lactate dehydrogenase, a recently discovered mammalian (including human) flavoenzyme. The paper unveils two main features: the enzyme performs best using manganese as metal cofactor and is active on several 2-hydroxy acid substrates, all sharing an aliphatic/hydrophobic side chain (i.e. the hydroxy equivalents of the branched-chain amino acids).

The work is relevant because it demonstrates that D-hydroxy acids can be a source of branch-chain alpha-keto acids and the three-dimensional structure nicely rationalizes the basis for the substrate selectivity of the enzyme. An additional point of interest is that the structural and biochemical analysis explains the pathogenicity of the mutations causing the enzyme deficiency, a genetic disease. The mutations negatively impact on the enzyme stability and affect residues that are directly or indirectly involved in catalysis. The other points made by the manuscript are mostly confirmatory. The catalytic mechanism is confirmed to involve a hydride transfer step whereas the FAD-binding site is confirmed to be similar to that of other flavoenzymes.

Methodologically, the work is sound and the crystallographic investigation is of the finest quality.

Response: We are very grateful for the positive comments about our work and the constructive suggestions for the revision.

Specific comments:

Comments 1: The manuscript is long and occasionally repetitive. I would suggest to shorten it to make it more focused on what is really new, and possibly unanticipated. For instance, the text extensively discusses the FAD binding site that hardly reveals anything new. The basis of the stereo-selectivity is the same as discussed before for other oxidases and dehydrogenases. The Discussion repeats the points that have been analysed and discussed in the Results. Better to focus the Discussion on what is new and has been learned from this work.

Response: We thank the reviewer for the constructive suggestion. To address this comment, in the revised manuscript, we have made the following changes:

- (1) We have removed the "Structure of the FAD-binding site" section, and instead added a brief discussion on the FAD binding with the protein to the end of the "Crystal structures of mLDHD" section (Page 11, paragraph 3). Accordingly, we have moved Fig. 3a to Supplementary Information as Supplementary Fig. 7.

- (2) We have shortened the discussion on the substrate stereo-selectivity (Page 14, paragraph 3).
- (3) We have shortened the first paragraph of the Discussion to make it more concise.
- (4) We have also shortened some repetitive parts/sentences of the paper to make them more concise.

Comment 2: The overall enzyme structure should be better compared to that of other enzymes. What are the most similar enzyme structures present in the PDB? What are the differences?

Response: We thank the reviewer for the constructive suggestion. The search of homologous proteins sharing similar structures with mLDHD in the Protein Data Bank (PDB) via DALI server (Holm, 2020) identified two best candidates: *A. woodii* LctD (PDB: 7QH2) (Kayastha et al., 2022) with a Z-score of 47.2 and human D2HGDH (PDB: 6LPN) (Yang et al., 2021) with a Z-score of 44.9. Although *A. woodii* LctD and human D2HGDH only share a sequence identity of 29% and 27% with mLDHD, respectively, they adopt similar overall structures consisting of an FAD-binding domain, a substrate-binding domain and a C-terminal domain. Superposition of mLDHD with *A. woodii* LctD and human D2HGDH reveals an RMSD of 1.7 Å (for 386 aligned C α atoms) and 1.6 Å (for 338 aligned C α atoms), respectively. Similar to mLDHD, LctD also contains a highly hydrophobic substrate-binding subsite B and exhibits high substrate specificity for D-lactate against L-lactate.

To address this comment, we have added the these comparison results in the Discussion as follows: “**The search of homologous proteins sharing similar structures with mLDHD in the Protein Data Bank (PDB) via DALI server identified two best candidates: *A. woodii* LctD (PDB: 7QH2) with a Z-score of 47.2 and human D2HGDH (PDB: 6LPN) with a Z-score of 44.9. Although *A. woodii* LctD and human D2HGDH share only a sequence identity of 29% and 27% with mLDHD, respectively, they adopt similar overall structures consisting of an FAD-binding domain, a substrate-binding domain and a C-terminal domain. Superposition of mLDHD with *A. woodii* LctD and human D2HGDH reveals an RMSD of 1.7 Å (for 386 aligned C α atoms) and 1.6 Å (for 338 aligned C α atoms), respectively. *A. woodii* LctD is a homolog of mLDHD, and thus it also contains a highly hydrophobic substrate-binding subsite B and exhibits high substrate specificity for D-lactate against L-lactate.**” (Page 18, paragraph 4)

On the other hand, the major difference between mLDHD and human D2HGDH occurs at the substrate-binding site. The substrate-binding subsite B of mLDHD is a large, hydrophobic pocket; in contrast, the substrate-binding subsite B of human D2HGDH is a deep, charged pocket. This difference dictates their differed substrate specificities. The difference between LDHD and D2HGDH and its impact on the substrate specificities of these enzymes have been discussed in the Discussion (Page 19, paragraph 2).

Comment 3: The manganese dependency is intriguing. Do our cells contain enough manganese to saturate the enzyme that binds the metal with a 50 microM K_d?

Response: We thank the reviewer for the thoughtful comment. In the original manuscript, the mLDHD activity was measured at a series of Mn^{2+} concentrations (0, 5, 10, 20, 50, 100, 200, 400, 800, and 1000 μM), showing that mLDHD exhibits optimal specific activity with the Mn^{2+} concentration in the range of 50-200 μM . Thus, we used 50 μM Mn^{2+} in all the activity assays without thinking of the physiological concentration of Mn^{2+} in cells.

In the revision, to evaluate the optimal Mn^{2+} concentration range more precisely, we measured the mLDHD activity at the Mn^{2+} concentrations of 30 μM and 40 μM as well. The results show that mLDHD exhibits about 40% of the optimal activity in the presence of 10 μM Mn^{2+} , about 50% of the optimal activity in the presence of 20 μM Mn^{2+} , and reaches the optimum activity in the presence of 30 μM Mn^{2+} (see figure below). In the revised manuscript, we have added the new data to Supplementary Fig. 3. As the Mn^{2+} concentration at 30 μM or 50 μM has no obvious effect on the activity, we decide not to repeat all the activity assays at 30 μM Mn^{2+} again.

Supplementary Fig 3. Specific activity of mLDHD towards D-LAC with varied Mn^{2+} concentrations.

In the human body, the concentration of Mn varies depending on tissue and cell types. Previous studies reported that under normal condition, Mn is mainly accumulated in the liver (1.32 mg/kg wet tissue), pancreas (1.17 mg/kg wet tissue), kidney cortex (0.98 mg/kg wet tissue), and brain (0.18-0.46 mg/kg wet tissue); however, the exact intracellular concentration of Mn was not determined in these studies (Krebs et al., 2014; Rahil-Khazen et al., 2002). Another study reported that the intracellular concentration of Mn in human brain cells is about 20-53 μM under physiological condition and 60-150 μM under pathological conditions (Bowman and Aschner, 2014). At such Mn concentrations, LDHD is able to exert the full catalytic activity.

Comment 4: Please avoid an excessive use of abbreviations. I found the manuscript often hard to read because of this. I do not see any reason for not using the standard nomenclature for the substrates and products.

Response: We thank the reviewer for the constructive suggestion. To address this comment, we have changed the abbreviations of small molecules to standard nomenclature throughout the manuscript. In order to keep the figures neat and clean, we used the abbreviations in the

figures and listed the standard nomenclature corresponding to the abbreviations in the figure captions.

Comment 5: Did the authors check if the enzyme works also with cytochrome c or other non-oxygen electron acceptors such as DCPIP? Can it work as a dehydrogenase?

Response: We thank the reviewer for the thoughtful comment. In the original manuscript, we performed enzymatic activity assay in a standard reaction solution containing 50 mM Tris (pH 7.4), 50 mM Tris-HCl (pH 7.4), 2 μg mLDHD, 50 μM MnCl_2 , 200 μM PMS, 100 μM DCIP, and 1 mM substrate. To address this comment, we have performed enzymatic activity assay of mLDHD in the standard reaction solution using DCPIP or cytochrome c to replace DCIP. The reduction of DCPIP or cytochrome c was measured by the change of absorbance at 600 nm (for DCPIP) or 550 nm (for cytochrome c), respectively. D-lactate (1 mM) was used as the substrate. The results show that similar to DCIP, DCPIP and cytochrome c can act as an electron acceptor in the reaction when PMS was used as an intermediate electron acceptor. Compared to DCIP, mLDHD exhibits about 80% activity in the presence of DCPIP ($0.816 \pm 0.18 \mu\text{mol}\cdot\text{min}^{-1}\cdot\text{mg}^{-1}$) or cytochrome c ($0.822 \pm 0.005 \mu\text{mol}\cdot\text{min}^{-1}\cdot\text{mg}^{-1}$) (Figure R1). When PMS was omitted from the reaction solution, the mLDHD activity was undetectable in the presence of DCPIP or cytochrome c. Similarly, the mLDHD activity was also undetectable only in the presence of DCIP.

Figure R1. mLDHD activities in the presence of DCIP, DCPIP or cytochrome c (Cty C).

Later on, we found that DCIP (2,6-dichloroindophenol) and DCPIP (2,6-dichlorophenol-indophenol) are actually the same chemical compound. We speculated that the small difference in the specific activity of mLDHD when DCIP was substituted with DCPIP might be resulted from the differences of purities and hydration states for chemicals purchased from different companies. In the initial experiments, we used DCIP purchased from Sangon Biotech (Cat. No. A600396) with a purity $\geq 97\%$. In the revision, we used DCPIP purchased from Leyan (Cat. No. 1258428) with a purity of 95%.

Based on the above results, we believe that in the *in vitro* activity assay, mLDHD functions as an oxidase using PMS and DCIP (or DCPIP) as electron carriers and oxygen as a hydrogen acceptor. However, we cannot exclude the possibility that mLDHD might also

work as a dehydrogenase using FAD as a hydrogen carrier in a coupled oxidation-reduction reaction *in vivo*. As these results are not relevant to the topic of this work, we decide not to include these results in the revision.

Reviewer #2

Overall comments:

In this manuscript the authors perform steady-state kinetic and structural characterization of mouse lactate dehydrogenase (mLDHD) and disease associated variants, showing it has specificity to D-lactate and a broad range of hydrophobic D-2-hydroxyacids. The X-ray crystal structures were determined with FAD, Mn²⁺ and a series of substrates or products. The most interesting aspect of the work is that the enzyme may play an important role in the metabolism of these hydroxyacids and the part that disease-associated mutations of mLDHD may play in the pathogenesis of D-lactic acidosis. Indeed, knowledge of the potential substrates and products may be useful diagnostically. However, there are some concerns about some of the kinetic analyses and interpretations of the authors. **(1)** It is not clear why the crystalline enzyme has activity in the unmetallated form (where they have treated with EDTA) and this raises the question of whether the divalent cation is a cofactor or an activator. Is the enzyme simply contaminated with Mn²⁺ leading to the observed activity or is Mn²⁺ unnecessary? **(2)** The authors kinetics do not take into account the occupancy of FAD so it is uncertain if the k_{cat} values are correct; the ratio of enzyme to FAD cofactor is needed to know the concentration of active enzyme. **(3)** The manuscript could use another pass for grammar and usage.

Response: We thank the reviewer for the positive comments about our work and the constructive suggestions for revision.

(1) During the purification process, we added 1 mM EDTA in the purification buffer and the storage buffer to remove any potential unknown divalent metal ions derived from the expression and purification processes. Crystals of wild-type (WT) mLDHD with bound FAD (mLDHD^{FAD}) were grown from drops containing the reservoir solution of 4.0 M sodium formate. Structure determination confirmed that there is an FAD but no metal ion bound at the active site. Crystals of WT mLDHD in complex with FAD, Mn²⁺ and D-lactate were obtained by soaking the WT mLDHD^{FAD} crystals in the crystallization solution supplemented with 50 mM MnCl₂ and 50 mM D-lactate for 15-30 mins. Soaking of the mLDHD^{FAD} crystals for longer time often yielded the product-bound or mixed substrate/product-bound structure. It is evident that the crystalline mLDHD has activity towards the substrates in the soaking solution as there was saturated Mn²⁺. Thus, we prepared a loss-of-function H405A mutant mLDHD (mLDHD_{H405A}), and determined the crystal structures of the mutant in FAD-bound form and in complexes with FAD and various substrates. In the initial submission, we described the crystallization experiments and the soaking conditions clearly (Page 26, paragraph 1). However, we were negligent in the description of the crystallization experiments in the “Crystal structures of mLDHD” section by omitting the information about addition of Mn²⁺ in the soaking

solution. This omission caused the confusion of the reviewer about the functional role of Mn^{2+} as a cofactor or an activator. We apologize for this negligence.

In the revised manuscript, we have modified the sentence in the “Crystal structures of mLDHD” section as follows: “Crystallization experiments show that soaking of the mLDHD^{FAD} crystals in crystallization solution supplemented with the substrates and $MnCl_2$ often yields the product-bound or mixed substrate/product-bound structures” (Page 9, paragraph 3)

Our biochemical data show that the EDTA-treated mLDHD exhibits no activity in the absence of metal ions and exhibits activity in the presence of divalent cations including Mn^{2+} , Co^{2+} , Ni^{2+} and Ca^{2+} (Fig 1a). Thus, the divalent cation serves as a cofactor but not an activator for mLDHD. To attend this comment, in the revised manuscript, we have changed the sentence “As expected, the purified mLDHD has no activity in the absence of any metal ions (Fig 1a).” to “As expected, the EDTA-treated mLDHD has no activity in the absence of metal ions (Fig 1a).” In addition, we have added a sentence as follows: “It is evident that the divalent cation serves as a cofactor but not an activator for mLDHD.” (Page 6, paragraph 2)

- (2) We apologize for not taking the FAD occupancy into account in the activity assay. In the revision, we have determined the occupancy of FAD according to the reviewer’s suggestion. We have quantified the concentration of FAD using absorbance at 450 nm (A450), and the concentration of mLDHD using absorbance at 280 nm (A280). The results show that the molar ratio of FAD:protein in the purified mLDHD protein is 0.73 ± 0.04 . As suggested by the reviewer, we have calculated the concentration of active enzyme in the reaction solution based on the FAD occupancy and made corrections of the activity and kinetic parameters of mLDHD throughout the manuscript accordingly.

In the revised manuscript, we have also added the above methods in the Methods section as follows: “The concentration of the mLDHD protein was determined using the coextinction coefficient of protein at A280 ($42,565 M^{-1}cm^{-1}$). The concentration of bound FAD in the mLDHD protein was determined using a standard curve measured at A450. The molar ratio of FAD:mLDHD in the WT mLDHD protein was determined to be 0.73 ± 0.04 . The activity and kinetic parameters of mLDHD were corrected according to the concentration of active enzyme in the reaction solution calculated based on the FAD occupancy.” (Page 24, paragraph 3)

In addition, we have added a sentence in the “Expression and purification of mouse LDHD” section as follows: “Quantification of the FAD and protein concentrations shows that the molar ratio of FAD:protein is 0.73 ± 0.04 in the wild-type (WT) mLDHD protein.” (Page 6, paragraph 1)

- (3) We apologize for making the mistakes in grammar and improper usage of English. In the revision, we have taken extreme cautions to correct the obvious mistakes. In addition, we have asked a colleague to proofread the manuscript and make proper modifications and corrections.

Specific comments:

Comment 1: The authors state (pg 6) "Later structure determination of mLDHD in apo form confirms that there is an FAD but no metal ion bound "The word apo here is incorrectly used as that describes cofactor-free enzyme and FAD is bound. This should be described as metal free or Mn²⁺ free here and throughout the manuscript.

Response: We thank the reviewer for pointing out this misuse. In the revised manuscript, we have changed "apo form" to "FAD-bound form" or "mLDHD^{FAD}" throughout the manuscript to refer to the structure of mLDHD in complex with FAD.

Comment 2: the rates and k_{cat} given do not support the claim of a high catalytic activity- the k_{cat}/K_M is $6 \times 10^4 \text{ M}^{-1} \text{ s}^{-1}$ which falls below the numbers expected for high catalytic activity of 10^7 - $10^9 \text{ M}^{-1} \text{ s}^{-1}$ (diffusion controlled) and below the average of most physiological enzymes $10^5 \text{ M}^{-1} \text{ s}^{-1}$ [see Biochemistry (2011) ,50, pp 4402–4410]. The authors should address this activity with comparisons to the literature to insure that it first with the physiological role hypothesized for this enzyme.

Response: We thank the reviewer for this thoughtful comment and constructive suggestion. Our biochemical data show that the turnover rates (k_{cat}) of mLDHD for different substrates are in the range of 20.5-80.0 min^{-1} (or 0.34-1.33 s^{-1}); the K_M values are in the range of 13.9-337 μM ; and the catalytic efficiencies (k_{cat}/K_M) are in the range of 0.117-3.99 $\text{min}^{-1}\mu\text{M}^{-1}$ (or 1.95×10^3 - $6.65 \times 10^4 \text{ s}^{-1}\text{M}^{-1}$) (Table 1).

According to the previous study (Bar-Even et al., 2011), the median k_{cat} value for all the enzymes surveyed is $\sim 10 \text{ s}^{-1}$, where most k_{cat} values ($\sim 60\%$) are in the range of ~ 1 - 100 s^{-1} ; the median K_M of all the enzymes surveyed is $\sim 100 \mu\text{M}$, where $\sim 60\%$ of the K_M values are in the range of ~ 10 - $1000 \mu\text{M}$; and the median catalytic efficiency (k_{cat}/K_M) of all the enzymes surveyed is $\sim 10^5 \text{ s}^{-1}\text{M}^{-1}$, where the catalytic efficiencies of most of the enzymes ($\sim 60\%$) lie in the range of 10^3 - $10^6 \text{ s}^{-1}\text{M}^{-1}$. It was also shown that enzymes associated with central metabolism (the main carbon and energy flow and the metabolism of amino acids, fatty acids and nucleotides) tend to have higher turnover rates and catalytic efficiencies than those associated with secondary metabolism (involved in the regulation of metabolites produced in specific cells or tissues, under specific circumstances and/or in relatively limited amounts). In comparison to these results, the k_{cat} and k_{cat}/K_M values of mLDHD are comparable to those of enzymes associated with secondary metabolism (median $k_{cat} = 2.5 \text{ s}^{-1}$ and $k_{cat}/K_M = 6.3 \times 10^4 \text{ s}^{-1}\text{M}^{-1}$) but lower than those of enzymes associated with central metabolism (median $k_{cat} = 79 \text{ s}^{-1}$ and $k_{cat}/K_M = 4.1 \times 10^5 \text{ s}^{-1}\text{M}^{-1}$) (Bar-Even et al., 2011). This is in agreement with the high tissue-specific expression of mLDHD (Drabkin et al., 2019) and the miniscule amount of D-lactate (and very likely other D-2-hydroxyacids) in the human body under normal physiological conditions (Ewaschuk et al., 2005). These results indicate that mLDHD is a secondary metabolic enzyme associated with regulation of metabolites produced in specific cells or tissues, under specific conditions and/or in small amounts.

To address this comment, in the revised manuscript, we have made the following changes:

(1) We have changed the phrase "exhibits high enzymatic activity" to "exhibits enzymatic activity" when describing mLDHD activities towards D-2-hydroxyacid substrates throughout the manuscript.

(2) We have added the following paragraph in the “mLDHD has activity for a broad range of D-2-hydroxyacids with hydrophobic moieties” section: “**Compared to other metabolic enzymes, the k_{cat} and k_{cat}/K_M values of mLDHD are comparable to those of secondary metabolic enzymes that are involved in the regulation of metabolites produced in specific cells or tissues, under specific circumstances and/or at relatively low levels, but are lower than those of central metabolic enzymes that are involved in the main carbon and energy flow and the metabolism of amino acids, fatty acids and nucleotides. This indicates that mLDHD is a secondary metabolic enzyme, which is in agreement with the high tissue-specific expression of mLDHD and the miniscule amount of D-lactate (and very likely other D-2-hydroxyacids) in the human body under normal physiological conditions.**” (Page 8, paragraph 4)

Comment 3: Page: 7- It would be much easier to follow the arguments about substrate specificity if the ChemDraw Figures in S4 had the kinetic parameters (velocity) from 1D beside each structure

Response: In the revised manuscript, we have added the specific activity beside the molecular structure of each substrate in Supplementary Fig 4. In Table 1, we have listed the kinetic parameters for each substrate.

Comment 4: Page: 8- The authors state “mLDHD exhibits comparable activities and kinetic parameters compared to the pure D isomer substrates with similar chemical structures, indicating that the presence of L isomers has no effect on the activity of mLDHD towards the D-isomers and implying that the L-isomers cannot bind to mLDHD” I assume that when the authors state this that they are comparing the corrected concentration for the racemic mixtures (2 fold lower than pure isomers). This should be clarified.

Response: The reviewer is right. For the racemic mixtures, the substrate concentration used in the experiment was doubled and the specific activity and kinetic parameters were calculated based on the corrected concentration of D-isomer (1/2 of the concentration of the racemic substrate). To address this comment, we have added a sentence in Fig. 1 caption as follows: “**For the racemic substrates, the substrate concentration used in the experiment was doubled and the specific activity was calculated based on the corrected concentration of D-isomer (1/2 of the concentration of racemic substrate).**”

In addition, we have added a sentence in the footnote of Table 1 as follows: “For the racemic mixture, the substrate concentration used in the experiment was doubled, and **the kinetic parameters were calculated based on the corrected concentration of D-isomer (1/2 of the concentration of racemic substrate).**”

Comment 5: Page: 9- "As mLDHD has high activity towards the substrates, soaking of the apo WT mLDHD crystals in crystallization solution supplemented with the substrates often yielded the product-bound or mixed substrate/product-bound structures" This is a confusing statement compared to results in Figure S3, which I think shows no activity in the absence of Mn²⁺. If the enzyme is active in the absence of Mn²⁺ then the ion is not a cofactor but an activator. Please clarify throughout the manuscript. Are other enzymes in the enzyme family or subfamily (ie. LDHD, D2HGDG and GlcD) dependent on divalent cation?

Response: Please see our response to the overall comments of this reviewer. Briefly, in the soaking experiments, the WT mLDHD^{FAD} crystals were soaked in crystallization solution supplemented with substrates and MnCl₂. Therefore, crystalline mLDHD can exhibit enzymatic activity in the soaking solution.

LDHD belongs to the 2-hydroxyacid dehydrogenase subfamily of the VAO/PCMH flavoprotein family (Ewing et al., 2017). There are three members in this subfamily: LDHD, D2HGDD and GlcD (Ewing et al., 2017). Our biochemical data show that the EDTA-treated mLDHD exhibits no activity in the absence of metal ions and exhibits moderate to weak activity in the presence of divalent cations including Mn²⁺, Co²⁺, Ni²⁺ and Ca²⁺ (Fig 1a). Thus, the divalent cation is essential for the catalytic reaction and acts as a cofactor rather than an activator for mLDHD. D2HGDD can utilize Zn²⁺, Co²⁺ or Mn²⁺ as a cofactor, with the highest activity in the presence of Zn²⁺ (Yang et al., 2021). GlcD is insensitive to metal chelators such as EDTA, sodium azide and phenanthroline, indicating that GlcD does not require a metal ion as a cofactor (Lord, 1972).

Comment 6: Can the authors comment if the poorly defined surface residues are still included in the structure?

Response: The poorly defined surface residues are not included in the final structures. The missing residues in the structure models are listed in Table 2.

To address this comment, we have added a sentence in the text: “**The disordered surface residues are omitted from the final structure models (Table 2).**” (Page 9, paragraph 4)

Comment 7: The ratio of FAD bound to protein should be quantified by measurement of FAD absorption (OD 450) and protein assay. The kcat values may need to be corrected in cases where FAD occupancy is low.

Response: Please see our response to the overall comments of this reviewer. In the revision, we determined the occupancy of FAD according to the reviewer’s suggestion. We quantified the concentration of FAD using absorbance at 450 nm (A₄₅₀, Fig R2), and the concentration of mLDHD using absorbance at 280 nm (A₂₈₀). For the purified WT mLDHD protein, the molar ratio of FAD:protein is about 0.73 ± 0.04. For mLDHD_{W351C} and mLDHD_{R347W} mutants, the molar ratio of FAD:protein is 0.44 ± 0.02 and 0.48 ± 0.008, respectively. As suggested by the reviewer, we have made corrections of the activity and kinetic parameters of WT mLDHD as well as mLDHD_{W351C} and mLDHD_{R347W} throughout the manuscript accordingly.

Figure R2. Standard curve of FAD measured by absorbance at 280 nm used to determine the FAD concentration in the mLDHD protein.

In the revised manuscript, we have added the following sentences to describe FAD occupancy in the purified mLDHD protein: “Quantification of the FAD and protein concentrations shows that the molar ratio of FAD:protein in the wild-type (WT) mLDHD protein is 0.73 ± 0.04 .” (Page 6, paragraph 1)

In the Methods section, we have added the following sentences to describe the quantification of the concentrations of FAD and mLDHD in the purified mLDHD protein: “The concentration of the mLDHD protein was determined using the coextinction coefficient of protein at A280 ($42,565 \text{ M}^{-1}\text{cm}^{-1}$). The concentration of bound FAD in the mLDHD protein was determined using a standard curve measured at A450. The molar ratio of FAD:mLDHD in the WT mLDHD protein was determined to be 0.73 ± 0.04 . The activity and kinetic parameters of mLDHD were corrected according to the concentration of active enzyme in the reaction solution calculated based on the FAD occupancy.” (Page 24, paragraph 3)

Comment 8: The authors state- “Although the DL-isomer mixtures of some substrates (DL-2-HV, DL-2-HH, DL-2-HO, DL-2-HIC, and DL-2-HMV) were used in the crystallization experiments, only the D-isomers are found to bind to the active site, further confirming that the L-isomers cannot bind to mLDHD.” This is not exactly true depending on the relative Kds of the two isomers, one may simply be competing out the other. Especially as a very low occupancy (<5%) the L-isomer may not be detected in the X-ray structure.

Response: Our biochemical data show that mLDHD exhibits enzymatic activity for D-lactate, but no activity for L-lactate (Fig. 1d). In addition, L-lactate (up to 1 mM) exhibits no inhibitory effect for mLDHD when D-lactate is used as substrate (Supplementary Fig. 5). In the substrate-bound structures of mLDHD and mLDHD_{H405A}, all the substrates are clearly defined in the electron density maps. Although we cannot rule out the possibility that the L-isomers may bind to mLDHD with very weak affinity which may not be detected in the X-ray structure, the potentially weak binding of the L-isomers shows no effects on the enzymatic activity of mLDHD and the binding of the D-isomers to the active site.

To address this comment, in the revised manuscript, we have rephrased the sentence as follows: “Although the DL-isomer mixtures of some substrates (DL-2-hydroxyvalerate, DL-2-hydroxyhexanoate, DL-2-hydroxyoctanoate, DL-2-hydroxyisocaproate, and DL-2-

hydroxy-3-methylvalerate) were used in the crystallization experiments, only the D-isomers are found to bind to the active site, **further confirming that mLDHD has high stereoselectivity for the D-isomers over the L-isomers.**" (Page 10, paragraph 2)

Comment 9: The authors present the proposed mechanism of the enzyme in Figure 3F but has this mechanism been proposed or differ from the mechanisms proposed for other subfamily members? This should be in the Discussion.

Response: The catalytic mechanism for other members of the VAO/PCMH flavoprotein family has been previously proposed (Cunane et al., 2000). Although these enzymes catalyze a diversity of substrates, they follow a similar mechanism: one or more polar residues act as Lewis base to abstract a proton from the hydroxyl group of the substrate, and then a hydride anion is transferred from the substrate to the N5 atom of FAD to form a flavin hydroquinone anion. Our structural and biochemical data support this mechanism. Nevertheless, some enzymes do not require a metal ion to stabilize the substrate binding, and different enzymes use differed residues to bind the metal ion and substrate and to act as the Lewis base. Therefore, we believe it is worthy to briefly describe the proposed catalytic mechanism for mLDHD in the Results section.

To address this comment, we have rephrased some sentences of this section to reflect this fact. In particular, we have changed the sentence "Based on the structural and biochemical data, we can propose the molecular mechanism for the catalytic reaction of mLDHD oxidizing D-2-hydroxyacids into 2-ketoacids" to "**Based on the previously proposed catalytic mechanism for other members of the VAO/PCMH flavoprotein family and our structural and biochemical data, we can propose the catalytic mechanism for mLDHD to oxidize D-2-hydroxyacid into 2-ketoacid as follows (Fig. 3e).**" (Page 15, paragraph 3)

Comment 10: Minor grammatical errors/typos, missing labels

Page: 3- Accumulation of excess L- and D-LAC in human body can cause a disease called lactic acidosis" should read "Accumulation of excess L- and D-LAC in the human body ..."

Page: 4- "other organic acids in urea and plasm" I thinks this should read "other organic acids in urea and plasma." This also occurs on other pages- please fix all occurrences of plasm to plasma

Page: 5- "...only insoluble inclusion body" should read "...only insoluble inclusion bodies"

Page 6 - which is agreement with the weak alkaline physiological environment in vivo" should read "which is in agreement with the weak alkaline ..."

Page: 10- "which comprises of two subdomains" should read "which is comprised of two..."

Page: 11- The substrate-binding site is located in opposite to the si face" should read "...located opposite to the ..."

Page: 12- In Fig 3C the length of the Mn-H405 bind is missing

Page: 13- In Fig 2E I can't tell if 2.3 corresponds to the bond to His405 or lactate- one is missing.

Page 13- “the substrate binding and posit the C2 atom of the substrate in precise position should read “the substrate binding and position the C2 atom of the substrate precisely...”

Page 17- The authors state “Residues Thr251 and Thr440 of mLDHD are both located near the FAD-binding site (Fig. 4B).” Thr251 is not shown in the figure.

Page: 20- “In addition, the yielding BCKA products might be involved in the regulation of the BCAA metabolic pathway”- should read “In addition, the yielded BCKA products might be involved in the regulation of the BCAA metabolic pathway”

Response: We apologize for making these mistakes. In the revised manuscript, we have incorporated the suggested changes. We have also carefully and thoroughly proofread the manuscript to check the grammar, typos, labels and references. In addition, we have asked a colleague to proofread the manuscript and make proper modifications and corrections.

References

- Bar-Even, A., Noor, E., Savir, Y., Liebermeister, W., Davidi, D., Tawfik, D.S., and Milo, R. (2011). The moderately efficient enzyme: Evolutionary and physicochemical trends shaping enzyme parameters. *Biochemistry* 50, 4402-4410.
- Bowman, A.B., and Aschner, M. (2014). Considerations on manganese (Mn) treatments for in vitro studies. *Neurotoxicology* 41, 141-142.
- Cunane, L.M., Chen, Z.W., Shamala, N., Mathews, F.S., Cronin, C.N., and McIntire, W.S. (2000). Structures of the flavocytochrome p-cresol methylhydroxylase and its enzyme-substrate complex: Gated substrate entry and proton relays support the proposed catalytic mechanism. *J Mol Biol* 295, 357-374.
- Drabkin, M., Yogev, Y., Zeller, L., Zarivach, R., Zalk, R., Halperin, D., Wormser, O., Gurevich, E., Landau, D., Kadir, R., *et al.* (2019). Hyperuricemia and gout caused by missense mutation in D-lactate dehydrogenase. *J Clin Invest* 129, 5163-5168.
- Ewaschuk, J.B., Naylor, J.M., and Zello, G.A. (2005). D-lactate in human and ruminant metabolism. *J Nutr* 135, 1619-1625.
- Ewing, T.A., Fraaije, M.W., Mattevi, A., and van Berkel, W.J.H. (2017). The VAO/PCMH flavoprotein family. *Arch Biochem Biophys* 632, 104-117.
- Holm, L. (2020). DALI and the persistence of protein shape. *Protein Sci* 29, 128-140.
- Kayastha, K., Katsyv, A., Himmrich, C., Welsch, S., Schuller, J.M., Ermler, U., and Müller, V. (2022). Structure-based electron-confurcation mechanism of the ldh-etfab complex. *eLife* 11, e77095.
- Krebs, N., Langkammer, C., Goessler, W., Ropele, S., Fazekas, F., Yen, K., and Scheurer, E. (2014). Assessment of trace elements in human brain using inductively coupled plasma mass spectrometry. *J Trace Elem Med Biol* 28, 1-7.
- Lord, J.M. (1972). Glycolate oxidoreductase in *Escherichia coli*. *Biochim Biophys Acta Bioenerg* 267, 227-237.
- Rahil-Khazen, R., Bolann, B.J., Myking, A., and Ulvik, R.J. (2002). Multi-element analysis of trace element levels in human autopsy tissues by using inductively coupled atomic emission spectrometry technique (ICP-AES). *J Trace Elem Med Biol* 16, 15-25.

Yang, J., Zhu, H., Zhang, T., and Ding, J. (2021). Structure, substrate specificity, and catalytic mechanism of human D-2-HGDH and insights into pathogenicity of disease-associated mutations. *Cell Discov* 7, 3.

REVIEWER COMMENTS

Reviewer #1 (Remarks to the Author):

The manuscript has been carefully revised by the Authors and all Reviewers' comments have been taken care of. However, the answer to my comment 5 raises an issue that should be addressed. The Authors indicate that no enzymatic activity can be detected in the absence of phenazine methosulfate (PMS). PMS is a redox active molecular that can react with many molecules, e.g. with NAD(P)H. In their answer to comment 5, the Authors state that "When PMS was omitted from the reaction solution, the mLDHD activity was undetectable in the presence of DCPIP or cytochrome c". How did the Authors check for activity in the absence of DCPIP? How did they detect PMS reduction? Or did they look at substrate hydroxy acid consumption? Did the Authors check whether there is an oxidase activity using a simple horseradish peroxidase/ampex red or similar HRP-coupled assays to monitor production of hydrogen peroxide (of course in the absence of PMS)? If there is sustained H₂O₂-producing activity and it is comparable to the activity measured with PMS/DCPIP, then the enzyme can operate as oxidase in agreement with figure 3e and associated text. This should then be made clear and experimentally demonstrated. If there is no oxidase activity, the enzyme should be described as a dehydrogenase that poorly reacts with oxygen. In this scenario, the physiological electron acceptor would remain unknown since cytochrome c functions as electron acceptor only in a PMS-mediated process. I would then suggest that the Authors test ascorbic acid whose reduction/oxidation can be detected by HPLC-MS, as potential electron acceptor in the absence of PMS. Ideally, one would also like to see if there is activity with coenzyme Q or menaquinone. This experiment would probably be more complex as it would require the detection of substrate consumption by analytical methods and might be less feasible. These experiments will potentially clarify the nature of the physiological electron acceptor in the mitochondria.

The present version of the manuscript is not sufficiently clear about this point.

Minor point

-Please correct R347C to R347W in Figures 4c panel and legend.

Reviewer #2 (Remarks to the Author):

The authors have addressed my concerns and improved the manuscript based on my and Reviewer 1's critiques.

Point-by-point Responses to Reviewers' Comments

Reviewer 1's comment

Comment 1:

The manuscript has been carefully revised by the Authors and all Reviewers' comments have been taken care of. However, the answer to my comment 5 raises an issue that should be addressed. The Authors indicate that no enzymatic activity can be detected in the absence of phenazine methosulfate (PMS). PMS is a redox active molecular that can react with many molecules, e.g. with NAD(P)H. In their answer to comment 5, the Authors state that "When PMS was omitted from the reaction solution, the mLDHD activity was undetectable in the presence of DCIP or cytochrome c". How did the Authors check for activity in the absence of DCIP? How did they detect PMS reduction? Or did they look at substrate hydroxy acid consumption? Did the Authors check whether there is an oxidase activity using a simple horseradish peroxidase/ampex red or similar HRP-coupled assays to monitor production of hydrogen peroxide (of course in the absence of PMS)? If there is sustained H₂O₂-producing activity and it is comparable to the activity measured with PMS/DCIP, then the enzyme can operate as oxidase in agreement with figure 3e and associated text. This should then be made clear and experimentally demonstrated. If there is no oxidase activity, the enzyme should be described as a dehydrogenase that poorly reacts with oxygen. In this scenario, the physiological electron acceptor would remain unknown since cytochrome c functions as electron acceptor only in a PMS-mediated process.

I would then suggest that the Authors test ascorbic acid whose reduction/oxidation can be detected by HPLC-MS, as potential electron acceptor in the absence of PMS. Ideally, one would also like to see if there is activity with coenzyme Q or menaquinone. This experiment would probably be more complex as it would require the detection of substrate consumption by analytical methods and might be less feasible. These experiments will potentially clarify the nature of the physiological electron acceptor in the mitochondria.

The present version of the manuscript is not sufficiently clear about this point.

Response: We appreciate the reviewer for taking precious time to re-evaluate our manuscript, and are very delighted to know that our revision has appropriately addressed the reviewer's previous concerns and comments. We are also very grateful to the reviewer for the additional thoughtful comments and constructive suggestions for revision.

First, we have to apologize for the imprecise statement in the response to Comment 5. The previous statement that "When PMS was omitted from the reaction solution, the mLDHD activity was undetectable in the presence of DCIP or cytochrome c" inaccurately conveyed our meaning. The correct statement should be "When PMS was omitted from the

reaction solution and DCIP or cytochrome c was used as the sole electron acceptor, the activity of mLDHD was not detectable at the reaction conditions."

In this study, we used the PMS-DCIP assay to determine the activity of mLDHD (Figure R1a). The PMS-DCIP assay is a well-established method widely used for activity assay of flavin-dependent oxidases and dehydrogenases such as succinate dehydrogenase¹, sarcosine dehydrogenase² and D-amino acid oxidase³. Previously, we also used the PMS-DCIP assay to determine the activity of D-2-HG dehydrogenase⁴. As pointed out by the reviewer, PMS is a redox active molecule that can react with many molecules; thus, it is commonly used as the primary electron acceptor for *in vitro* kinetic studies of oxidases and dehydrogenases to ensure that full enzymatic activity would be measured^{1,5}. DCIP was used as the secondary electron acceptor and a redox indicator. In the catalytic reaction, mLDHD catalyzes the oxidation of substrate and the reduction of FAD. Then, the reduced FAD was oxidized by PMS and subsequently the reduced PMS was re-oxidized by DCIP. The enzymatic activity of mLDHD was determined by spectrophotometrically monitoring the reduction of DCIP at 600 nm for 3 min (extinct coefficient = 22.0 mM⁻¹cm⁻¹) (Figure R1a). Detailed method of the PMS-DCIP assay and the reaction conditions have been described in the Materials and Methods section.

In the PMS-cytochrome c (Cytc) assay, DCIP was replaced with cytochrome c as the secondary acceptor and redox indicator (Figure R1b). The enzymatic activity was measured by monitoring the reduction of cytochrome c at 550 nm for 3 min (extinct coefficient = 21.8 mM⁻¹cm⁻¹).

Figure R1. Reaction schemes of different assay systems used for determination of the LDHD activity. a. Reaction scheme of the PMS-DCIP assay. **b.** Reaction scheme of the PMS-cytochrome c (Cytc) assay.

In both assays, we did not measure PMS reduction or substrate consumption directly. Instead, the reduction of DCIP and Cytc were measured as the read-out of the catalytic reaction because DCIP and Cytc display significant absorption changes between the oxidized and reduced forms. When PMS was omitted from the reaction solution, the reduction of DCIP or Cytc was not detectable at the reaction conditions as the electron transfer chain was broken.

The reviewer suggested us to check the oxidase activity using the HRP-Amplex red assay. To address this comment, we carried out the HRP-Amplex red assay to measure H₂O₂ production in the absence of PMS (Figure R2a). D-lactate (1 mM) was used as the substrate. Enzymatic activity was measured by monitoring the absorption change of Amplex red at 570 nm for 20 min. A standard curve plotting varied concentrations of H₂O₂ versus the corresponding absorption changes of Amplex red was used to determine the amount of H₂O₂ produced in the reaction (Figure R2b). The assay results showed that mLDHD exhibited a V_{\max} of $0.108 \pm 0.001 \mu\text{mol}\cdot\text{min}^{-1}\cdot\text{mg}^{-1}$ (Figure R2c), which was approximately 9% of the V_{\max} as determined using the PMS-DCIP assay ($V_{\max} = 1.22 \pm 0.01 \mu\text{mol}\cdot\text{min}^{-1}\cdot\text{mg}^{-1}$, Table 1). This indicates that mLDHD could utilize O₂ as an electron acceptor but with a low catalytic efficiency at the reaction conditions.

Figure R2. LDHD activity determined by the HRP-Amplex red assay. **a.** Reaction scheme of the HRP-Amplex red assay used for determination of the LDHD activity. **b.** The standard curve plotting varied H₂O₂ concentrations versus corresponding absorption changes of Amplex red. **c.** The saturation curve of mLDHD determined by the HRP-Amplex red assay. The reaction solution consists of 50 mM Tris (pH 7.4), 2 μg mLDHD, 50 μM MnCl₂, 5 U/mL HRP, 100 μM Amplex red, and 0-1 mM D-lactate. The activity of mLDHD was measured by monitoring the absorption change of Amplex red at 570 nm for 20 min. All data points represent mean \pm SEM of three independent experiments.

The reviewer also suggested us to test whether ascorbic acid or other substances (such as coenzyme Q and menaquinone) could function as potential physiological electron acceptor in the absence of PMS. Since dehydroascorbic acid (DHAA) is the oxidized form of ascorbic acid, we used DHAA as the primary electron acceptor, which will be converted to ascorbic acid by reduced FAD in the reaction (Figure R3a). Previous studies showed that DCIP can be used as a redox indicator for ascorbic acid production^{6,7}. Thus, we set up the activity assay system using DHAA as the primary electron acceptor and DCIP as the secondary electron acceptor and redox indicator (Figure R3a). D-lactate (1 mM) was used as the substrate. The reduction of DCIP was measured spectrophotometrically at 600 nm. The assay results showed that when the reaction took place for 3 min, the change of DCIP absorption was insignificant; however, when the reaction took place for 20 min, the change of DCIP absorption was significant and the mLDHD exhibited an apparent specific activity of $0.011 \pm 0.002 \mu\text{mol}\cdot\text{min}^{-1}\cdot\text{mg}^{-1}$, which was about 1% of the specific activity as determined using the PMS-DCIP assay or 10% of the specific activity as determined using the HRP-Amplex red assay (Figure R3b). These results indicate that DHAA is a low-efficient electron acceptor compared to PMS or O₂ at the reaction conditions (Figure R3b).

Figure R3. Determination of the LDHD activity using various electron acceptors.

a. Reaction scheme of the DHAA-DCIP assay used for determination of the LDHD activity. **b.** Apparent specific activity of mLDHD determined using different assay methods with different electron acceptors. **c.** Reaction scheme of the DCIP assay used to measure the mLDHD activity. **d.** Reaction scheme of the Cyt_c assay used to measure the mLDHD activity. In all assays, the reaction solution contained 50 mM Tris (pH 7.4), 2 μ g mLDHD, 50 μ M MnCl₂, and 1 mM D-lactate, and different electron acceptors/redox indicators and reaction time were applied as follows. The PMS-DCIP assay: 200 μ M PMS and 100 μ M DCIP, 3 min; the PMS-Cyt_c assay: 200 μ M PMS and 100 μ M Cyt_c, 3 min; the HRP-Amplex red assay: 5 U/mL HRP and 100 μ M amplex red, 20 min; the DHAA-DCIP assay: 200 μ M DHAA and 100 μ M DCIP, 20 min; the DCIP assay: 100 μ M DCIP, 20 min; and the Cyt_c assay: 100 μ M Cyt_c, 20 min. All assays were performed in three independent experiments.

In the above experiments of the DHAA-DCIP assay, we realized that when the efficiency of the primary electron acceptor (DHAA) was low, a longer reaction time was required to observe significant absorption change of the redox indicator (DCIP). In the previous revision process, we observed that when PMS was omitted from the reaction solution, the reduction of DCIP or Cyt_c could not be detected based on a 3-min reaction. To examine whether DCIP or Cyt_c could also act as weak electron acceptor, we performed the DCIP assay (Figure R3c) and Cyt_c assay (Figure R3d) for a longer reaction time (20 min). This time, we could observe the reduction of DCIP or Cyt_c, indicating that mLDHD displayed weak activity under these conditions. The assay results showed that mLDHD exhibited an apparent specific activity of $0.008 \pm 0.001 \mu\text{mol}\cdot\text{min}^{-1}\cdot\text{mg}^{-1}$ for DCIP, and $0.028 \pm 0.001 \mu\text{mol}\cdot\text{min}^{-1}\cdot\text{mg}^{-1}$ for Cyt_c (Figure R3b). These results indicate that similar to DHAA, DCIP and Cyt_c could also act as low-efficient electron acceptors for mLDHD. Notably, as the apparent specific activities determined by the DHAA-DCIP assay, the DCIP assay and the Cyt_c assay were lower than that determined by the HRP-Amplex red assay, we could not exclude the possibility that O₂ may compete with these low-efficient electron acceptors in the catalytic reactions.

In summary, our *in vitro* activity assay results showed that mLDHD exhibited differed activity using the PMS-DCIP, DHAA-DCIP, DCIP and Cyt_c assays. In particular, mLDHD displayed the highest activity when PMS was used as the primary electron acceptor. Compared to PMS, Cyt_c, DHAA and DCIP are relatively low-efficient electron acceptors for the catalytic reaction by mLDHD. These results indicate that mLDHD could function as a dehydrogenase using PMS, Cyt_c, DHAA or DCIP as electron acceptor. On the other hand, mLDHD also displayed weak activity using the HRP-Amplex red assay, indicating that mLDHD could also function as an oxidase using molecular oxygen as the oxidant. As the

primary electron acceptor for the catalytic reaction by mLDHD under physiological conditions is unknown, we could not clarify whether mLDHD functions as an oxidase or a dehydrogenase *in vivo*.

Exploring the exact electron acceptor for the catalytic reaction by mLDHD *in vitro* and *in vivo* will undoubtedly help to further our understanding of the function and catalytic mechanism of mLDHD; however, it would require more complex experiments and more thorough studies to achieve this goal. As this study focuses on the discovery of mLDHD functioning as a general dehydrogenase for a broad range of D-2-hydroxyacids containing hydrophobic moieties, the molecular basis for the broad substrate specificity of LDHD and the functional roles of mutations in the pathogenesis of D-lactic acidosis, we would prefer not to make a thorough investigation into and a detailed discussion about whether mLDHD would function as an oxidase or a dehydrogenase. Thus, we decided not to include the above preliminary biochemical results about the determination of the mLDHD activity using different assay methods in the manuscript.

To attend the reviewer's comment, we have modified the scheme for the catalytic mechanism of mLDHD (**Figure 3e**), in which we removed O₂ and H₂O₂ and added “oxidant” to represent an unknown electron acceptor in the final step of the catalytic reaction. Accordingly, we have re-phrased the description of the catalytic reaction in the text as follows: “Subsequently, the 2-ketoacid product is dissociated from the active site, and the reduced FAD is oxidized to FAD by an oxidant. In our *in vitro* activity assay, phenazine methosulfate (PMS) was used as the primary oxidant followed by 2,6-dichloroindophenol (DCIP). However, the primary oxidant under physiological conditions is unknown, and thus it remains unclear whether mLDHD would function as an oxidase when molecular oxygen is used as the oxidant or a dehydrogenase when other electron acceptor(s) are used as the oxidant in a coupled oxidation-reduction reaction.” (Page 15, paragraph 2)

Figure 3e. Catalytic mechanism of mLDHD.

Comment 2:

Minor point

-Please correct R347C to R347W in Figures 4c panel and legend.

Response: We thank the Reviewer for pointing out the mistake. We have changed R347C to R347W in **Figure 4c** and the figure legend.

Reviewer 2's comment

The authors have addressed my concerns and improved the manuscript based on my and Reviewer 1's critiques.

Response: We appreciate the reviewer for taking precious time to re-evaluate our manuscript. We are very delighted to know that our revision has appropriately addressed the reviewer's concerns and comments, and by doing so, the quality of the manuscript has been improved.

References

1. Ackrell, B.A.C., Kearney, E.B. & Singer, T.P. Mammalian succinate dehydrogenase. in *Methods in Enzymology*, 53, 466-483 (Academic Press, 1978).
2. Cook, R.J. & Wagner, C. Dimethylglycine dehydrogenase and sarcosine dehydrogenase: Mitochondrial folate-binding proteins from rat liver. in *Methods in Enzymology*, 122, 255-260 (Academic Press, 1986).
3. Sacchi, S., Pollegioni, L., Pilone, M.S. & Rossetti, C. Determination of D-amino acids using a D-amino acid oxidase biosensor with spectrophotometric and potentiometric detection. *Biotechnology Techniques* **12**, 149-153 (1998).
4. Yang, J., Zhu, H., Zhang, T. & Ding, J. Structure, substrate specificity, and catalytic mechanism of human D-2-HGDH and insights into pathogenicity of disease-associated mutations. *Cell Discov* **7**, 3 (2021).
5. Jahn, B., *et al.* Understanding the chemistry of the artificial electron acceptors PES, PMS, DCPIP and Wurster's Blue in methanol dehydrogenase assays. *J Biol Inorg Chem* **25**, 199-212 (2020).
6. Karayannis, M.I. Kinetic determination of ascorbic acid by the 2,6-dichlorophenolindophenol reaction with a stopped-flow technique. *Anal Chim Acta* **76**, 121-130 (1975).
7. Dabrowski, K., Lackner, R. & Doblender, C. Ascorbate-2-sulfate sulfohydrolase in fish and mammal. Comparative characterization and possible involvement in ascorbate metabolism. *Comp Biochem Physiol B* **104**, 717-722 (1993).

REVIEWERS' COMMENTS

Reviewer #1 (Remarks to the Author):

The Authors have carefully and convincingly addressed the point about the electron acceptor.